# The causes and consequences of Alzheimer's disease: phenome-wide evidence from Mendelian randomization

Roxanna Korologou-Linden [1,2] ✉, Laxmi Bhatta[3], Ben M. Brumpton [3,4,5], Laura D. Howe[1,2], Louise A. C. Millard[1,2,6], Katarina Kolaric[1,2], Yoav Ben-Shlomo[2], Dylan M. Williams[7,8], George Davey Smith [1,2], Emma L. Anderson[1,2,9], Evie Stergiakouli [1,2,9] & Neil M. Davies [1,2,3,9]

Alzheimer's disease (AD) has no proven causal and modifiable risk factors, or effective interventions. We report a phenome-wide association study (PheWAS) of genetic liability for AD in 334,968 participants of the UK Biobank study, stratified by age. We also examined the effects of AD genetic liability on previously implicated risk factors. We replicated these analyses in the HUNT study. PheWAS hits and previously implicated risk factors were followed up in a Mendelian randomization (MR) framework to identify the causal effect of each risk factor on AD risk. A higher genetic liability for AD was associated with medical history and cognitive, lifestyle, physical and blood-based measures as early as 39 years of age. These effects were largely driven by the *APOE* gene. The follow-up MR analyses were primarily null, implying that most of these associations are likely to be a consequence of prodromal disease or selection bias, rather than the risk factor causing the disease.

Late-onset Alzheimer's disease is an irreversible neurodegenerative disorder which accounts for the majority of dementia cases[1]. Despite major private and public investments in research, there are no effective treatments for preventing the disease[2]. Many risk factors and biomarkers have been identified to be associated with the risk of Alzheimer's disease[3]. Most treatments (99.6%) developed to halt Alzheimer's disease failed in phase I, II, or III trials[4]. One explanation for these failures is that the identified risk factors and drug targets are a consequence of Alzheimer's disease rather than underlying causes.

Genetic epidemiologic methods, such as Mendelian randomization (MR), can potentially provide more reliable insights into the causal mechanisms underlying the associations between risk factors and disease[5]. To date, hypothesis-driven MR studies have found mixed evidence for a causal role of cardiovascular risk factors in the development of Alzheimer's disease[6–8].

Phenome-wide association studies (PheWAS) are a hypothesis-free method, similar to genome-wide association studies (GWAS), which estimate the associations between a genotype or polygenic risk score (PRS) and the phenome[9]. PheWAS can potentially elucidate the phenotypic consequences of Alzheimer's disease, and critically when in the life course, these effects emerge.

Here, we estimated the associations of genetic liability for Alzheimer's disease and the phenome by age to identify the earliest effects of the disease process (i.e., genetic liability to Alzheimer's disease as a risk factor). We then tested whether the identified variables were a cause or a consequence of Alzheimer's disease using two-

[1]Medical Research Council Integrative Epidemiology Unit, Bristol Medical School, University of Bristol, Bristol BS8 2BN, UK. [2]Population Health Sciences, Bristol Medical School, University of Bristol, Barley House, Oakfield Grove, Bristol BS8 2BN, UK. [3]K.G. Jebsen Center for Genetic Epidemiology, Department of Public Health and Nursing, NTNU, Norwegian University of Science and Technology, Trondheim, Norway. [4]Clinic of Medicine, St. Olavs Hospital, Trondheim University Hospital, Trondheim, Norway. [5]HUNT Research Center, Department of Public Health and Nursing, NTNU, Norwegian University of Science and Technology, Levanger, Norway. [6]Intelligent Systems Laboratory, Department of Computer Science, University of Bristol, Bristol, UK. [7]MRC Unit for Lifelong Health and Ageing at UCL, University College London, London, UK. [8]Department of Medical Epidemiology & Biostatistics, Karolinska Institutet, Stockholm, Sweden. [9]These authors contributed equally: Emma L. Anderson, Evie Stergiakouli, Neil M. Davies. ✉e-mail: r.korologou-linden@bristol.ac.uk

sample MR (i.e., phenome associated with Alzheimer's disease genetic liability as the exposure).

# Results

## Sample characteristics

The UK Biobank sample is 55% female (39–53 years, mean = 47.2 years, standard deviation (SD) = 3.8 years) in tertile 1, 55% female (53–62 years, mean = 58.03 years, SD = 2.4 years) in tertile 2 and 49% female (62–72 years, mean = 65.3 years, SD = 2.7 years) in tertile 3. In the whole sample, the Alzheimer's disease PRS was associated with a lower age at recruitment (β: −0.006 years; 95% confidence interval (CI): −0.01, −0.0002; $P = 0.007$). The mean standardized PRS (95% CI) in each tertile was as follows: 0.006 (−0.0003, 0.01); and 0.001 (−0.01, 0.009) and −0.007 (−0.02, 0.002) ($P$ for trend = 0.01).

## Association of Alzheimer's disease polygenic risk score and the phenome

We examined the effects of genetic liability to Alzheimer's disease on the UK Biobank phenome, using 18 single-nucleotide polymorphisms (SNPs), robustly and independently associated with Alzheimer's disease ($P \le 5 \times 10^{-8}$) (Fig. 1A). Phenome-wide association analyses were performed within each age tertile of UK Biobank. The age tertiles of UK Biobank are tertile 1, ages 39–53 years; tertile 2, ages 53–62 years and tertile 3, 62–72 years. Each tertile consisted of 111,656 participants. Selected PheWAS hits are presented in Figs. 2–4 and Supplementary Figs. 3–5. Results for continuous outcomes are in terms of a 1 SD change of inverse rank normal transformed outcome and log-odds or odds for binary or categorical outcomes. A higher PRS for Alzheimer's disease was associated with own diagnosis and family history of dementia, diagnoses of cardiovascular diseases, and self-reported history of high cholesterol and pure hypercholesterolemia. Furthermore, participants with a higher PRS had an increased risk of using cholesterol-lowering drugs, in addition to beta-blockers and aspirin in all age tertiles (Supplementary Fig. 3). A higher PRS was associated with lower body mass index and various body fat measures, lower diastolic blood pressure and higher spherical power in the oldest participants (i.e., the strength of lens needed to correct focus) (Fig. 2). In addition, on average, participants with higher PRS performed worse and took longer to complete cognitive tests in all age tertiles (39–72 years) (Fig. 3). Participants with a higher PRS also had a higher weighted-mean mode of anisotropy in the left inferior fronto-occipital fasciculus for participants aged 53–72 years (Fig. 3). There was evidence of an association between higher PRS and blood cell composition markers and these associations increased with age (Fig. 4). On average, the parents of participants with a higher PRS for Alzheimer's disease died at a younger age (Supplementary Fig. 4). There was strong evidence that a higher PRS was associated with healthier dietary choices (Supplementary Fig. 4) and lifestyles (e.g., frequent exercise) in the two oldest tertiles (ages 53–72 years) (Supplementary Fig. 5). For previously implicated factors in Alzheimer's disease, a higher PRS was associated with higher systolic blood pressure only in participants aged 39–53 years and higher pulse pressure for all age ranges. There was some evidence of an association between the PRS and a lower number of pack-years of smoking for the oldest participants (Supplementary Fig. 5).

**Sensitivity analysis.** We replicated the top PheWAS hits from the oldest age tertile (i.e., where most associations were observed) in the Nord-Trøndelag Health Study (HUNT). In participants aged 62–72 years, of the 165 variables identified in the UK Biobank PheWAS, we replicated 32 variables with adequate precision for the age-stratified analysis, 20 of which were directionally consistent and replicated at $P \le 0.05$. The effects of genetic liability to Alzheimer's disease on blood-based biomarkers and physical measures in HUNT closely mirror those in UK Biobank (Figs. 5 and 6). Other replicated effects

included a lower odds of the participant's mother having diabetes, dietary habits such as a higher oily fish intake and fresh fruit intake, and lifestyle habits such as frequent sleeplessness/insomnia (Supplementary Figs. 7–8, 10). Supplementary Figs. 6–10 show forest plots for all measures. We repeated the analysis, estimating the associations of the PRS and the 21,849 variables in UK Biobank for the entire sample (there are additional phenotypes due to the higher occurrence of events in all participants). We identified the effects of Alzheimer's disease genetic liability on the variables detected in the age-stratified analysis with larger precision and additional variables (Supplementary Figs. 11–14). For example, in the analysis using the entire sample, a higher PRS was associated with metabolic dysfunction phenotypes such as diabetes and obesity (Supplementary Fig. 11); a higher volume of gray matter in the right and left intracalcarine and supracalcarine cortices (Supplementary Fig. 13); and additional blood-based biomarkers (Supplementary Fig. 13).

When we repeated the analysis by removing SNPs tagging the *APOE* region from the PRS using the whole sample, we could not replicate most of the hits detected in the oldest tertile. The non-*APOE* PRS was associated with higher odds of own and family diagnosis of Alzheimer's disease (Supplementary Fig. 15) and lower odds of family history of chronic bronchitis/emphysema (Supplementary Fig. 16). There was evidence that the non-*APOE* PRS was associated with worse performance in cognitive tests (Supplementary Fig. 17).

## Two-sample Mendelian randomization of UK Biobank phenotypes on Alzheimer's disease

We performed a two-sample MR of the identified PheWAS variables on the risk of Alzheimer's disease (i.e., the reverse model to PheWAS). The strongest associations following correction for multiple testing are shown in Fig. 7. We found evidence that a one SD higher genetically predicted whole body fat-free and water mass decreased the risk of Alzheimer's disease (OR: 0.78; 95% CI: 0.69, 0.88 and OR: 0.81; 95% CI: 0.71, 0.91, respectively). A one SD higher genetically predicted forced vital capacity resulted in 22% lower odds of risk for Alzheimer's disease (OR: 0.78; 95% CI: 0.67, 0.91) (Supplementary Table 12). Furthermore, a higher genetically predicted basal metabolic rate was protective for Alzheimer's disease risk (OR: 0.75; 95% CI: 0.66, 0.85). We observed that a higher genetic liability of doing more moderate physical activity (at least 10 minutes) but not self-reported vigorous activity increased the odds of developing Alzheimer's disease (OR: 2.69; 95% CI:1.39, 5.17 and OR: 1.02; 95% CI: 0.33, 3.18, respectively) (Supplementary Table 16). For previously implicated risk factors for Alzheimer's disease, we found a higher genetic liability for having a college degree and Alevel qualifications reduced the risk of Alzheimer's disease (Supplementary Table 17).

**Assessing pleiotropy.** To evaluate the validity of the MR assumptions, we quantified pleiotropy using heterogeneity statistics and pleiotropy-robust methods (e.g., MR Egger regression). Due to the large sample size of the exposures, the instrument strength of all SNPs was relatively high ($F > 30$). However, the SNPs used for each body measurement implied highly heterogeneous effects on the risk of Alzheimer's disease (all heterogeneity Q statistic $P < 1.99 \times 10^{-5}$). This heterogeneity may be due to horizontal pleiotropy (Supplementary Figs. 19–25). The causal effect estimates for forced vital capacity and basal metabolic rate were also heterogenous (Supplementary Table 12 and Supplementary Figs. 19 and 20).

**Assessing causal direction.** We used Steiger filtering to evaluate the direction of causation between Alzheimer's disease and the phenotypes identified in the PheWAS[10]. We found little evidence that SNPs explained more variance in the outcome than the exposure for most results reported in the two-sample MR section (Supplementary Data 7–18).

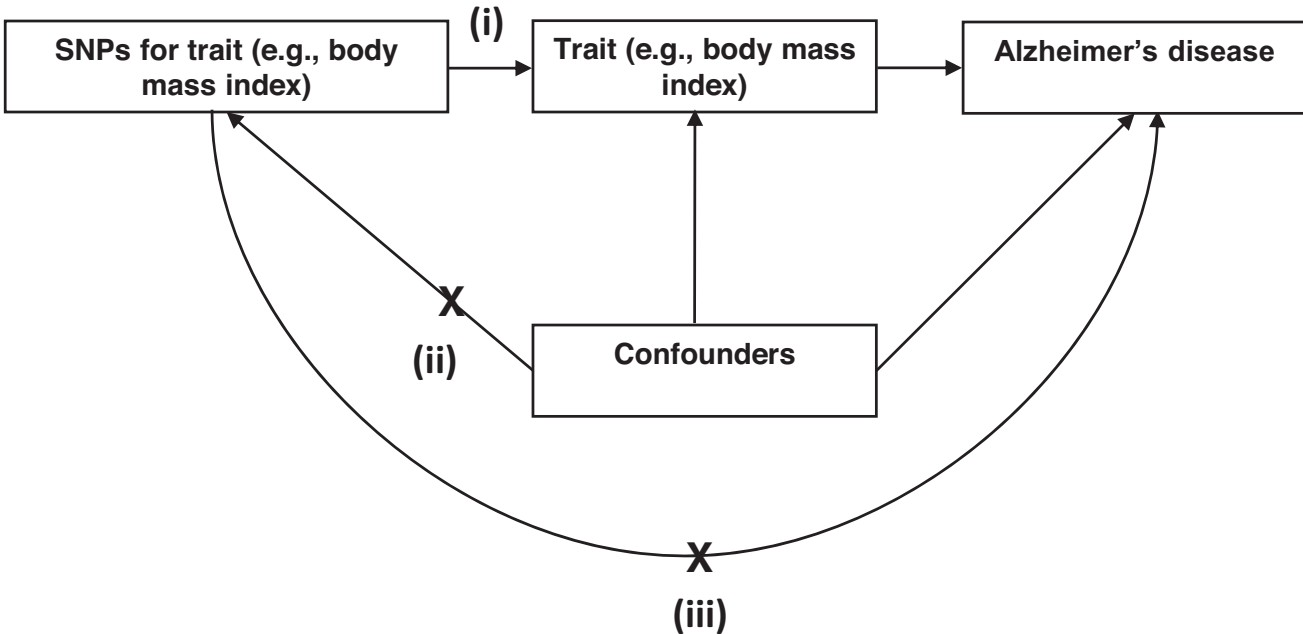

**A. Phenome-wide association study**

**B. Two sample Mendelian randomization**

**Fig. 1 | Study design for the phenome-wide association study of Alzheimer's disease genetic liability and follow-up Mendelian randomization of identified phenotypes on Alzheimer's disease.** Diagram (**A**) describes our study design when conducting a phenome-wide association study, and diagram (**B**) describes our study design when using Mendelian randomization. In **A**, the polygenic risk score for Alzheimer's disease may either have a downstream causal effect on the trait (e.g., body mass index), or it may affect the trait through pathways other than through Alzheimer's disease (i.e., pleiotropic effects). Diagram (**B**) describes our follow-up analysis using Mendelian randomization to establish the causality and directionality of the observed associations. In **B**, we test the hypothesis that the trait (e.g., body mass index) causally affects liability to Alzheimer's disease, provided that the conditions (i), (ii), and (iii) are satisfied. The polygenic risk score for the trait of interest is a valid instrument in that (i) the single nucleotide polymorphisms for a trait are strongly associated with the trait they proxy (relevance), (ii) there are no confounders of the single nucleotide-outcome relationship (independence), and (iii) the single nucleotide polymorphisms only affect the outcome via their effects on the trait of interest (exclusion restriction).

## Discussion

In our study, we conducted a hypothesis-free phenome-wide scan to investigate how and at what age the Alzheimer's disease PRS affects the phenome. The effects of a higher genetic liability for Alzheimer's disease are stronger in participants of aged 62–72 years, although the direction of effect is similar across age groups. We investigated whether the effects observed are likely to be causes or consequences of the disease process using two-sample bidirectional MR. We found evidence that a minority of the variables identified in the PheWAS are likely to causally affect the liability to Alzheimer's disease. Of the variables associated with Alzheimer's disease genetic liability in the PheWAS, these included basal metabolic rate, forced vital capacity, whole body fat-free and whole body water mass, and self-reported moderate physical activity. Of the factors previously implicated in Alzheimer's disease risk, these included A level/AS qualifications and college degree qualifications (Fig. 7).

The PheWAS suggested that increased genetic liability for Alzheimer's disease affected a diverse array of phenotypes such as medical history, brain-related phenotypes and physical, lifestyle and blood-based measures. However, these effects appear to be primarily driven by variation in the *APOE* gene. Our sensitivity analysis, excluding the *APOE* region, replicated only the effects for family history of Alzheimer's disease and some cognitive measures. These findings are in line with a recent study which showed that a higher PRS excluding *APOE* was particularly deleterious for the age of AD onset in *APOE*ε4 carriers, with no evidence of such an effect in *APOE*ε4 non-carriers[11]. Furthermore, these results are consistent with observational studies[12–17] and studies in *APOE*-deficient mice[18–21], which demonstrate the multifunctional role of *APOE* on longevity-related phenotypes such as changes in lipoprotein profiles[18,22,23], neurological disorders[24], type II diabete[19], altered immune response[20], and increased markers of oxidative stress[21]. Therefore, this strongly suggests that the effects of

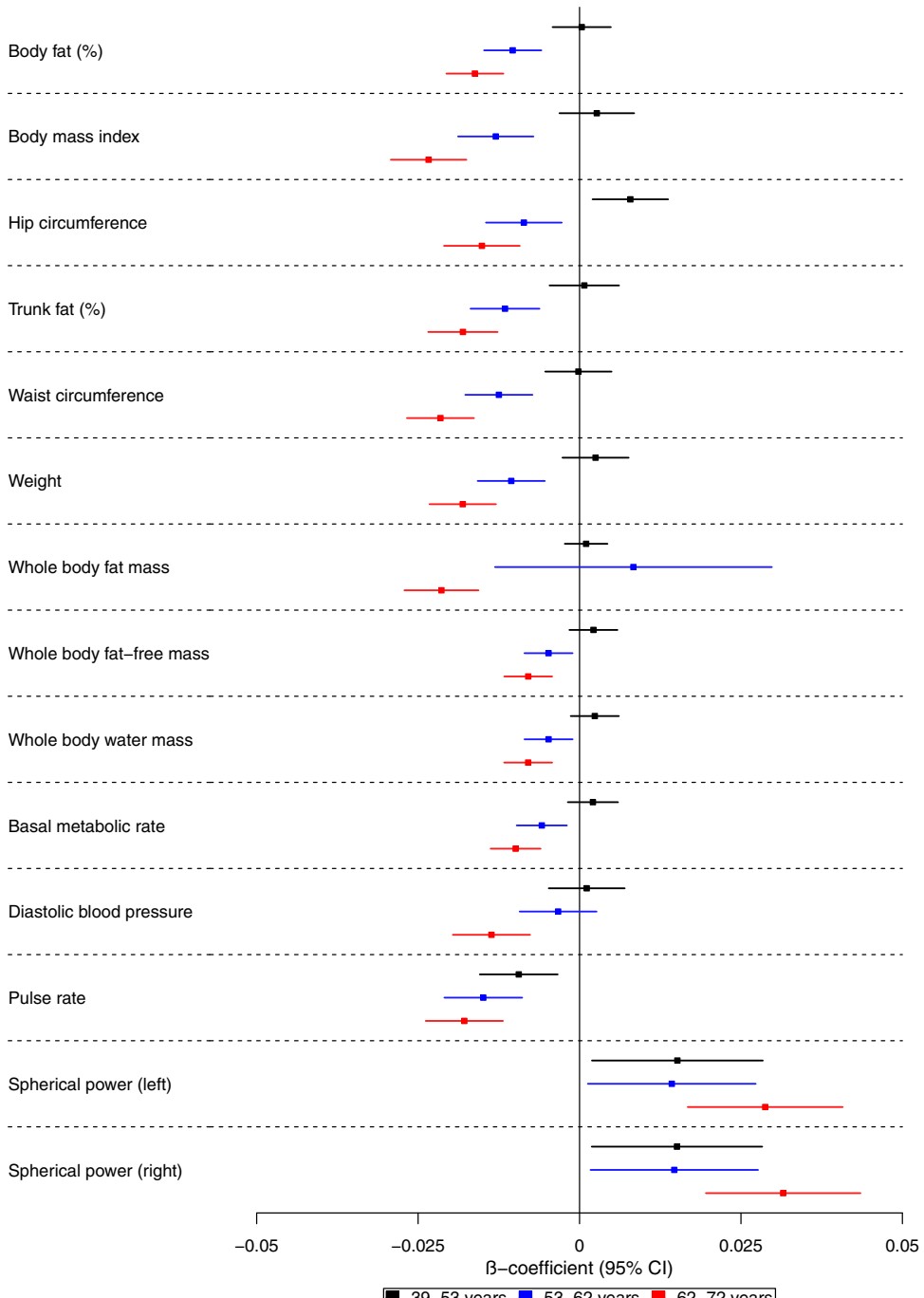

**Fig. 2 | Forest plot showing the effect estimates for the association between the polygenic risk score for Alzheimer's disease (including the *APOE* region) and physical measures.** Legends at the bottom of each graph indicate age tertiles. Each tertile consists of 111,656 participants, ordered by age. Effect estimates represent an SD change in the phenotype per 1 unit increase in the standardized polygenic risk score for Alzheimer's disease. Error bars represent 95% confidence intervals. All statistical tests were two-sided. There is evidence that the polygenic risk score for Alzheimer's disease is related to physical measures in older but not younger participants. This suggests that Alzheimer's disease causes these changes rather than vice versa.

Alzheimer's PRS on the phenome (e.g., atherosclerotic heart disease) are likely to be due to biological pathways related to *APOE*.

Previous observational studies have reported conflicting evidence on the association of cardiovascular risk factors with Alzheimer's disease, including hypertension. The results depend on the age at which these risk factors were measured[25–27]. Similarly, in our study, a higher PRS for Alzheimer's disease was associated with lower body mass index and body fat in participants of ages 53-72 years and lower diastolic blood pressure in the participants of ages 62-72 years. In agreement with some previous MR studies[6,7,28,29], we found little evidence that body mass index and blood pressure causally affect the risk of developing Alzheimer's disease. Hence, the association observed in the PheWAS between the PRS, lower body fat measures, and diastolic blood pressure is likely to reflect the prodromal disease process. We found evidence that a higher self-reported number of days of moderate physical activity increased the odds of Alzheimer's disease. A study[30] found directionally consistent effects with the results of our analysis, but the evidence that moderate to vigorous physical activity affected the risk of Alzheimer's disease was weak.

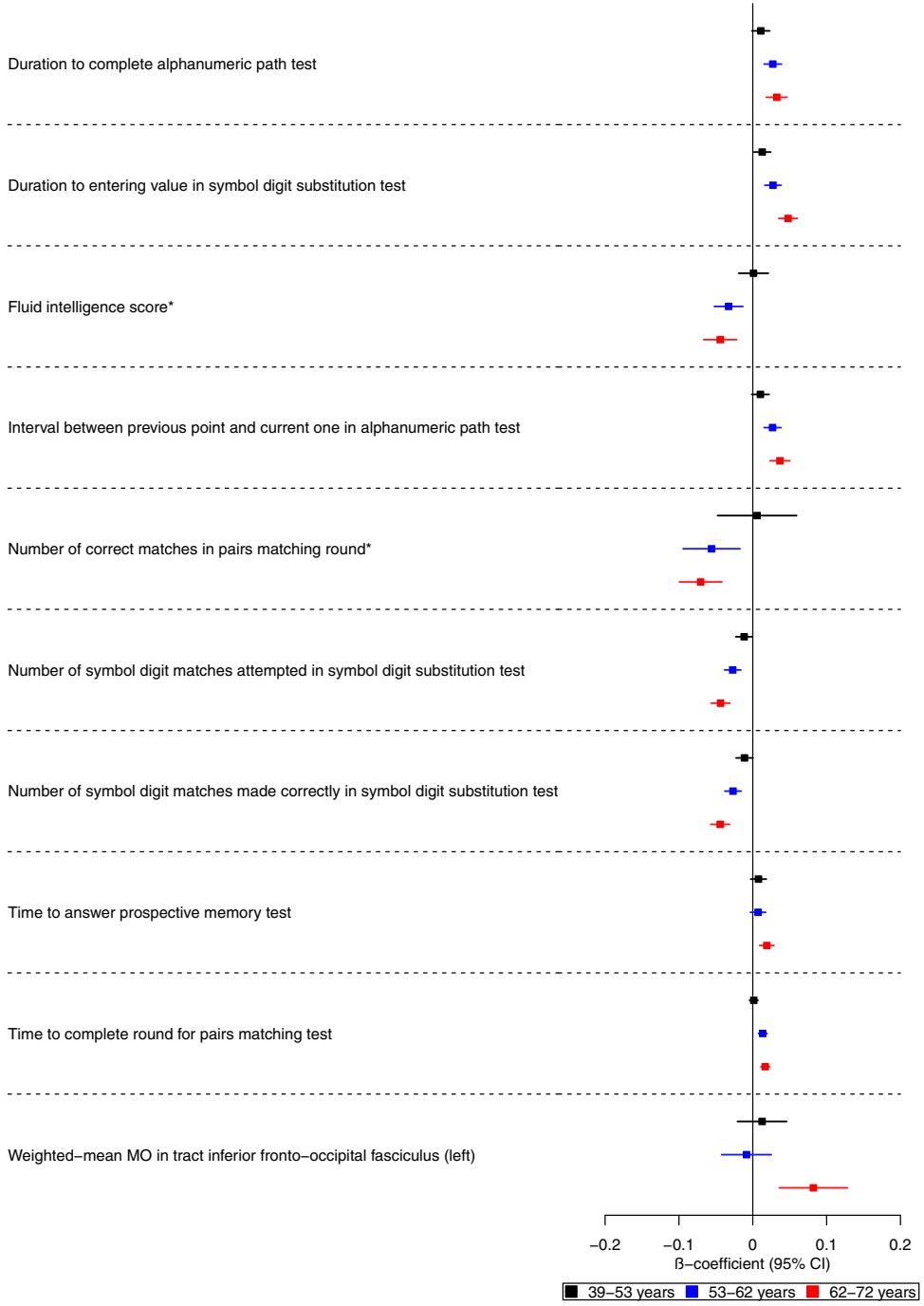

**Fig. 3 | Forest plot showing the effect estimates for the association between the polygenic score for Alzheimer's disease (including the *APOE* region), cognitive and brain-related measures.** Legends at the bottom of each graph indicate age tertiles. Each tertile consists of 111,656 participants, ordered by age. Effect estimates represent an SD change in the phenotype per 1 unit increase in the standardized polygenic risk score for Alzheimer's disease. Error bars represent 95% confidence intervals. All statistical tests were two-sided. *Effect estimates were derived from ordered logistic models, and effect estimates are on the log-odds scale. We found evidence that the polygenic risk score for Alzheimer's disease is related to some cognitive measures in all age ranges examined. This may suggest a bidirectional relationship between cognitive measures and Alzheimer's disease.

We found the genetic liability for Alzheimer's disease associated with several variables involving inflammatory pathways such as self-reported wheeze/whistling, monocyte count, and blood-based measures. This agrees with previous evidence of genetic correlations between Alzheimer's disease and asthma[31] and longitudinal studies[32,33]. In our study, red blood cell indices show the earliest evidence of association with the genetic liability of Alzheimer's disease. Still, we found little evidence that these measures caused Alzheimer's disease using MR, indicating that cell composition changes may be an early consequence of Alzheimer's disease pathophysiology. Previous studies found genetic variants associated with red blood cell distribution width are linked to autoimmune disease and Alzheimer's disease[34,35]. Once the function of the specific variants involved in specific biological pathways related to Alzheimer's disease is further elucidated, pathway-specific PRS may be used to inform the causality of the biological phenotypes identified in our PheWAS, such as blood-based biomarkers. We found evidence that a PRS for Alzheimer's disease was associated with a lower fluid intelligence score, as previously reported[36] but not educational

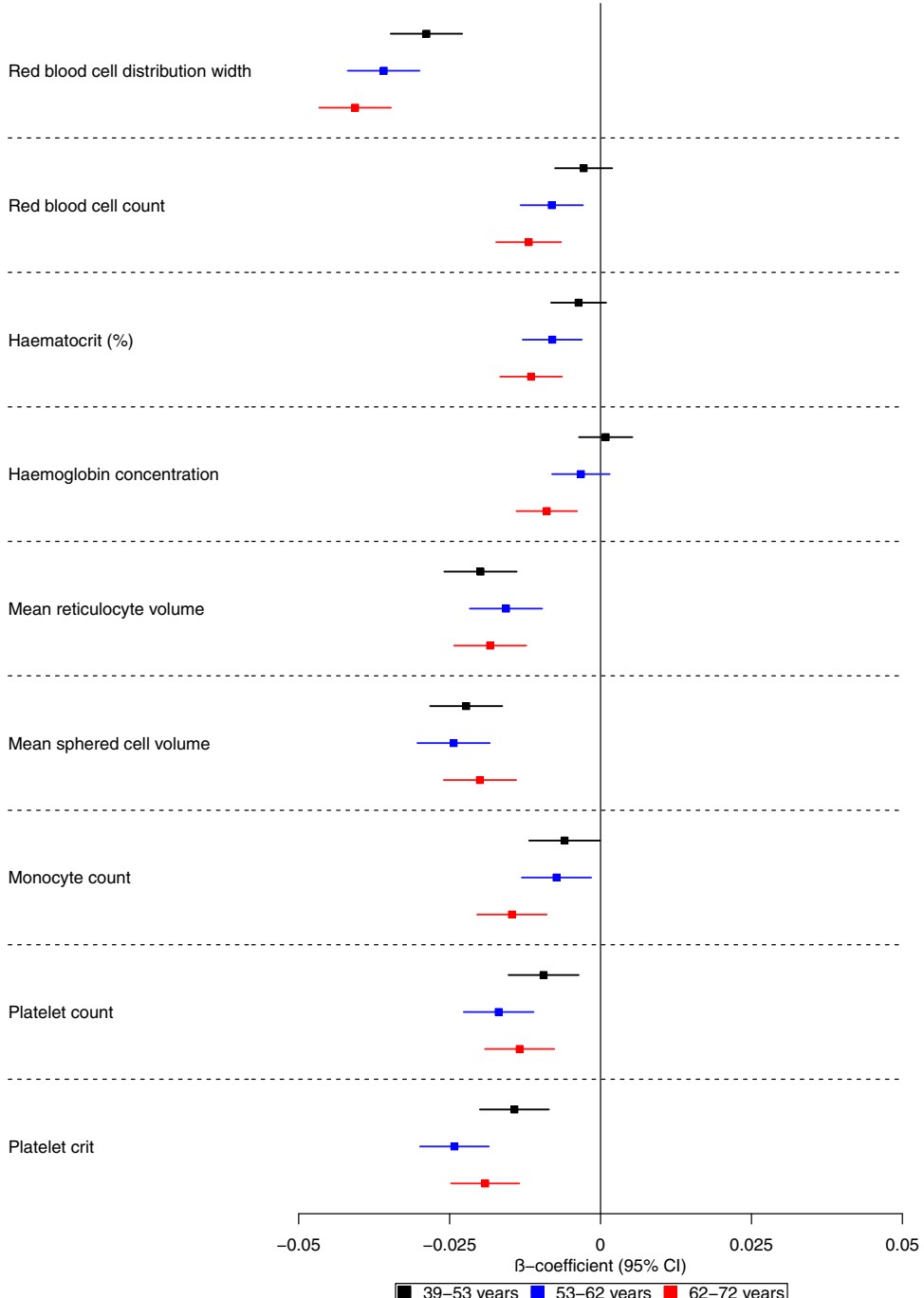

**Fig. 4 | Forest plot showing the effect estimates for the association between the polygenic score for Alzheimer's disease (including the *APOE* region) and biological measures.** Legends on the bottom of each graph indicate age tertiles. Each tertile consists of 111,656 participants, ordered by age. Effect estimates represent an SD change in the phenotype per 1 unit increase in the standardized polygenic risk score for Alzheimer's disease. Error bars represent 95% confidence intervals. All statistical tests were two-sided. The UK Biobank included an age-dependent increase in the effect of the polygenic risk score on blood-based measures. This may indicate that blood-based markers may be causal in the development of Alzheimer's disease.

attainment. Although previous MR studies have suggested that higher educational attainment reduces liability for Alzheimer's disease[6,37,38], a multivariable MR study found little evidence that educational attainment directly increased the risk of Alzheimer's disease over and above the underlying effects of intelligence[39].

Our study shows that the most likely reason for the contrast between the primarily null MR findings in our study, and the vast evidence base for Alzheimer's disease risk factors in observational studies, is that none of the factors identified to be associated with Alzheimer's disease in such studies are likely to be causal. Instead,

they are either symptoms of prodromal Alzheimer's disease (i.e., biased by reverse causation) or spurious associations due to confounding and/or selection bias. This conclusion is corroborated by the findings observed in the PheWAS, which are similar to those in observational studies (such as body mass index and sleep disturbances). In addition, our study's lack of positive findings was not due to insufficient statistical power, as the effects were precisely estimated. Unlike PRS and association studies, MR aims to disentangle causality from correlation. Observational studies are prone to biases such as confounding and reverse causation. On the

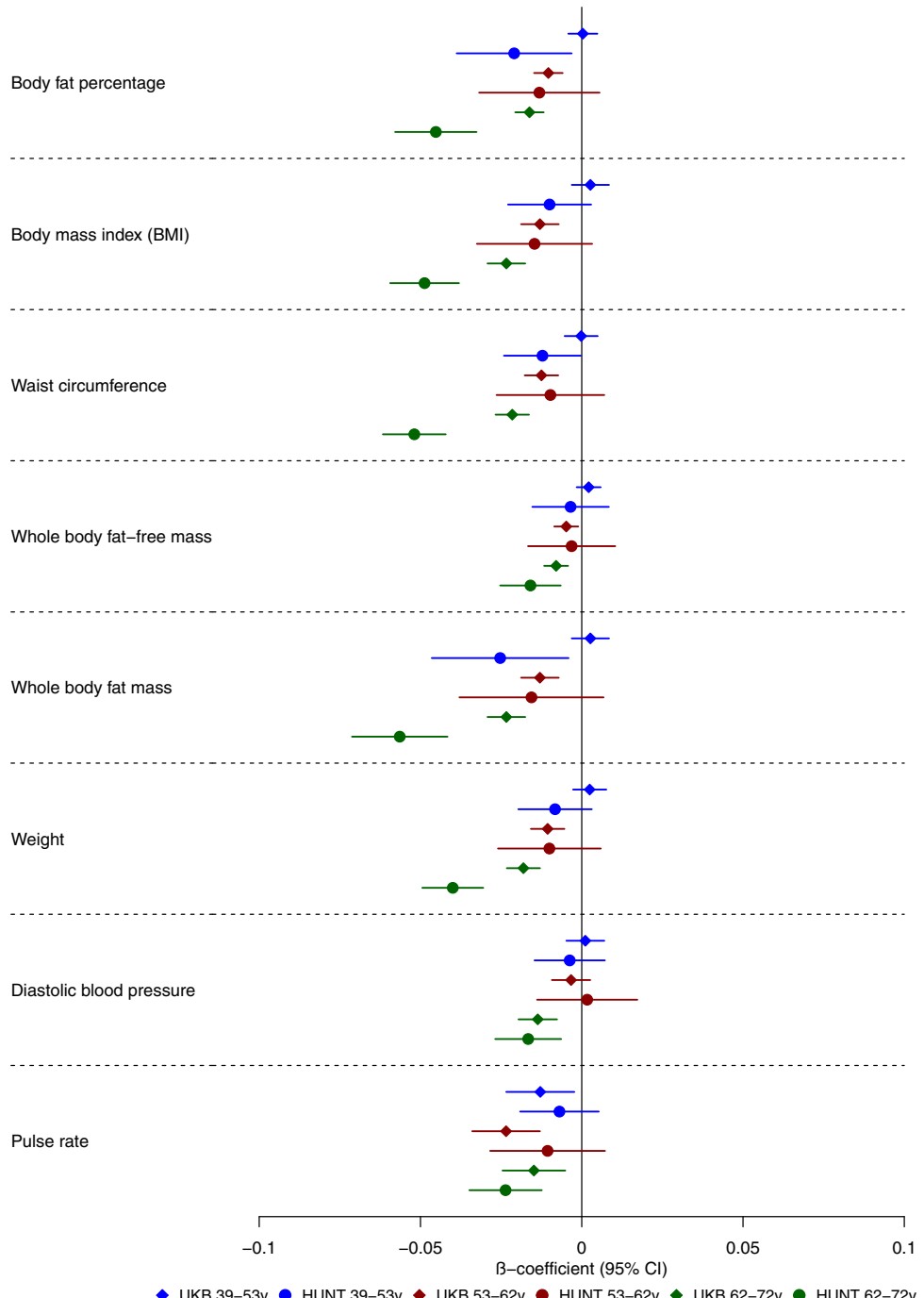

**Fig. 5 | Forest plot showing the age-stratified effect estimates for the association between the polygenic score for Alzheimer's disease (including the *APOE* region) and physical measures in UK Biobank (diamond markers) and HUNT (replication sample, circle markers).** Legends at the bottom of each graph indicate age tertiles in both cohorts. The colors represent blue, the youngest age tertile (39–53 years); red; middle age tertile (53–62 years); green, the oldest age tertile (62–72 years). Effect estimates represent an SD change in the phenotype per 1 unit increase in the standardized polygenic risk score for Alzheimer's disease. Error bars represent 95% confidence intervals. All statistical tests were two-sided. The UK Biobank analyses included 9,043–111,485 participants in each tertile. The HUNT analyses included 7,267–18,307 participants in each age tertile. The confidence intervals are smaller in UK Biobank due to the larger sample size of the cohort compared to HUNT.

contrary, as MR uses genetic variants as proxies for the tested exposures, the possibility of confounding and measurement error is negligible[5]. Hence, our study highlights that most of the risk factors identified in observational literature are likely a response to Alzheimer's disease pathological processes.

The large sample of UK Biobank provided unparalleled statistical power to investigate the phenotypic manifestation of a higher genetic liability for Alzheimer's disease by age group. Furthermore, the

systematic approach of searching for effects using PheWAS reduces the bias associated with hypothesis-driven investigations.

The Alzheimer's disease PRS may have horizontal pleiotropic effects on different traits and disorders, which may result in heterogeneity. MR and instrumental variable estimators for binary outcomes also cannot assume a constant treatment effect; as such it is expected that the MR estimates if sufficiently powered, would be heterogeneous[40]. With the available data, it is impossible to

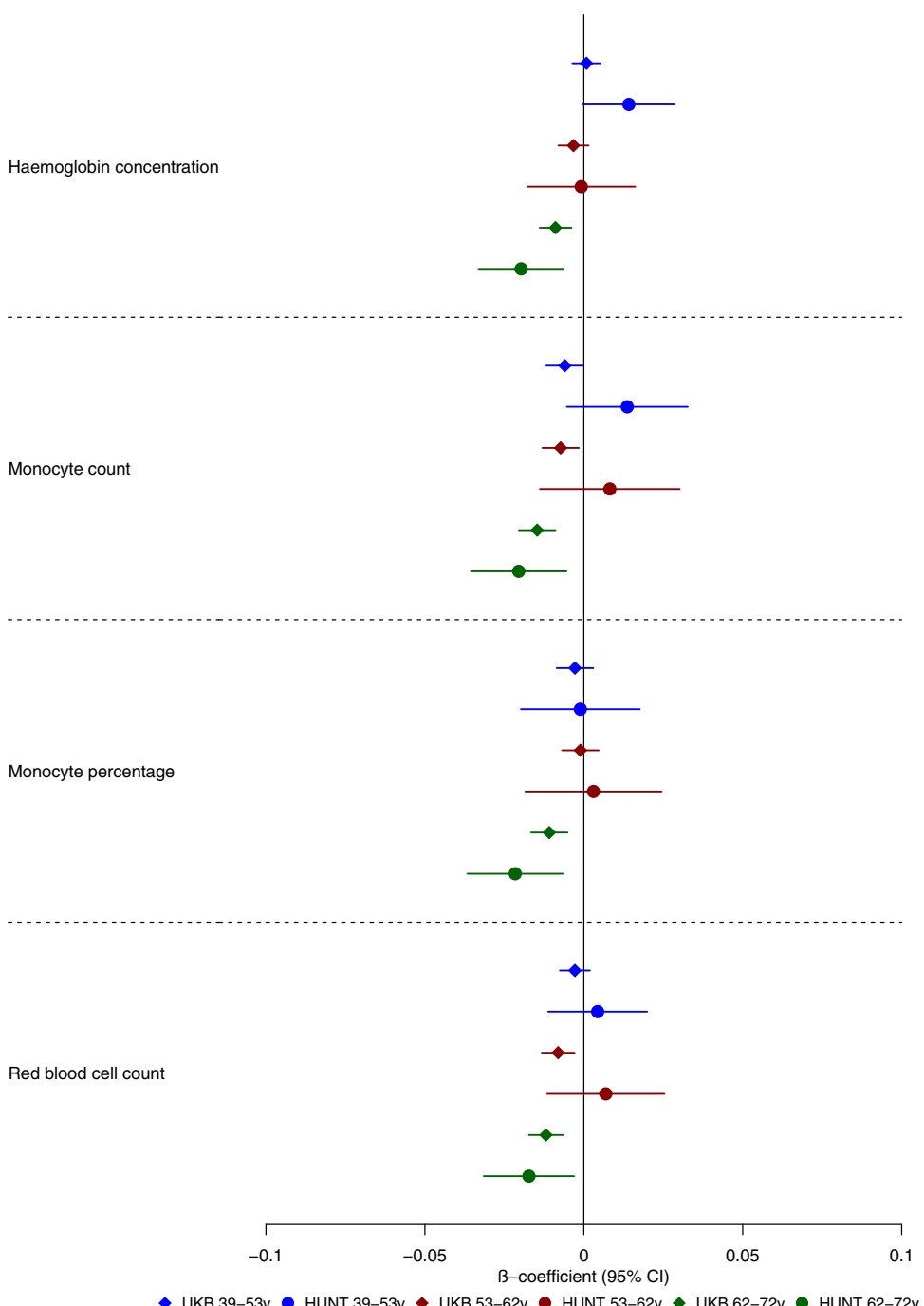

**Fig. 6 | Forest plot showing the age-stratified effect estimates for the association between the polygenic score for Alzheimer's disease (including the *APOE* region) and blood-based biomarker measures in UK Biobank (diamond markers) and HUNT (replication sample, circle markers).** Legends at the bottom of each graph indicate age tertiles in both cohorts. The colors represent blue, the youngest age tertile (39–53 years); red; middle age tertile (53–62 years); green, the oldest age tertile (62–72 years). Effect estimates represent an SD change in the phenotype per 1 unit increase in the standardized polygenic risk score for Alzheimer's disease. Error bars represent 95% confidence intervals. All statistical tests were two-sided. The HUNT analyses included 7,267–18,307 participants for these measures. The UK Biobank analyses included 108,095–108,391 participants for blood-based biomarkers. The confidence intervals are smaller in UK Biobank due to the larger sample size of the cohort compared to HUNT.

determine whether this heterogeneity is due to pleiotropy or heterogeneous treatment effects. The results from both the PheWAS and the MR follow-up could be explained by collider bias, which may have been introduced into our study by selecting the study sample. The UK Biobank includes a highly selected, healthier sample of the UK population[41]. Compared to the general population, participants are less likely to be obese, smoke, drink alcohol daily, and have fewer self-reported medical conditions[42]. Selection bias may occur if those

with a lower genetic liability to Alzheimer's disease and a specific trait (e.g., higher education or higher levels of physical activity) are more likely to participate in the study. This could induce an association between genetic liability for Alzheimer's disease and the traits in our study[43]. Furthermore, if both the PRS for Alzheimer's disease and the examined traits associate with survival, sampling only living people can induce spurious associations that do not exist in the general population[44,45]. Such bias may have affected our findings for body

| Phenotype | PRS with *APOE* | PRS without *APOE* | Reverse MR |
|---|---|---|---|
| **PheWAS** | | | |
| Basal metabolic rate | - | X | - |
| Forced vital capacity | X | X | - |
| Whole body fat-free mass | - | X | - |
| Whole body water mass | - | X | - |
| Self-reported moderate-to-vigorous physical activity | + | X | + |
| **Previously implicated risk factors** | | | |
| A levels/AS levels | X | X | - |
| College degree | X | X | - |

**Fig. 7 | Association of Alzheimer's disease polygenic risk score with the phenome, and estimated effect of each phenotype using Mendelian randomization.** These findings showed evidence of association in the MR framework, following a correction for multiple testing using a strategy controlling for the false discovery rate. + and − indicate the direction of the coefficient for phenotypes associated with Alzheimer's disease using two-sample MR. X represents associations that were consistent with the null.

mass and physical activity, as individuals with a higher body mass index or infrequent physical activity and those with higher values of the Alzheimer's PRS are less likely to survive and participate in UK Biobank. The PRS for Alzheimer's disease in our analysis was associated with lower age at recruitment, suggesting that older people with higher score values are less likely to participate. Hence, considering these limitations, the variables that may be associated with selection or survival[41] should be interpreted with caution. In this PheWAS, we identified that a higher genetic liability for Alzheimer's disease is associated with 165 of the 15,402 UK Biobank variables. MR analysis follow-up showed evidence that only seven of these factors were implicated in the etiology of Alzheimer's disease. We found little evidence that the remaining phenotypes examined are likely to modify the disease process, but the association with the Alzheimer's disease PRS is likely due to reverse causation or selection bias. Further research should exploit the full array of potential relationships between the genetic variants implicated in Alzheimer's disease, intermediate phenotypes, and clinical phenotypes by using other omics and phenotypic data to identify possible biological pathways changing the risk of Alzheimer's disease.

## Methods
### Study design
Our analysis proceeded in two steps. First, we ran a PheWAS of the Alzheimer's disease PRS and all available variables in the UK Biobank, stratifying the sample by age. Second, we followed up all variables associated with the PRS in a bidirectional MR analysis. We outline the research questions answered by the PheWAS and the MR approach in Fig. 1.

### Sample description
UK Biobank is a population-based study of 503,325 people recruited between 2006 and 2010 from across Great Britain[46,47]. This work was done under application number 16729 (version 2 genetic data [500 K with HRC imputation] and phenotype dataset 21753). In Supplementary Fig. 1, the flowchart shows the number of participants removed at each stage of the quality control pipeline. The UK Biobank study resource has ethical approval and its own ethics committee (https://www.ukbiobank.ac.uk/learn-more-about-uk-biobank/governance/ethics-advisory-committee). A full description of the study design, participants, and quality control (QC) methods has been published[48,49]. Briefly, participants were excluded due to familial relatedness and non-Caucasian ancestry. A total sample of 334,968 remained after QC (Supplementary Fig. 1).

### Polygenic risk score
We constructed a standardized weighted PRS including single-nucleotide polymorphisms (SNPs) associated with Alzheimer's disease at $P \leq 5 \times 10^{-8}$ for UK Biobank participants, based on the summary statistics from a meta-analysis of the International Genomics of Alzheimer's Project (IGAP)[50], the Alzheimer's Disease Sequencing Project (ADSP)[51] and the Psychiatric Genetics Consortium (PGC)[52] (24,087 cases and 55,058 controls). SNPs were clumped using $r^2 > 0.001$ and a physical distance for clumping of 10,000 kb. A polygenic risk score including 18 genetic variants was calculated for each participant with genetic data using PLINK (version 1.9). Each score was calculated from the effect size (logarithm (log) odds)-the weighted sum of 18 alleles associated with Alzheimer's disease within each participant. Our primary analysis used the PRS, including variants near the *APOE* gene (Chr 19: 44,400–46,500 kb)[53]. The *APOE* region explains a large proportion of the variance in the polygenic risk score ($R^2 = 84\%$). The PRS was standardized by subtracting the mean and dividing it by the standard deviation (SD) of the PRS.

### Main analysis
The full UK Biobank sample was divided into three age-stratified sub-samples ($n = 111,656$ in each tertile) with the aim of examining the age-varying effects of the PRS for Alzheimer's disease. We performed

PheWAS within each tertile. Age, sex, and the first ten genetic principal components were included as covariates.

## Outcomes

The Biobank data showcase enables researchers to identify variables based on the field type (http://biobank.ctsu.ox.ac.uk/showcase/list.cgi). There were 2655 fields of the following types: integer, continuous, and categorical (single and multiple). We excluded 55 fields a priori, including age and sex, and technical variables (e.g., assessment center) (Supplementary Table 2).

## Statistical analyses

**Phenome-wide association study.** We estimated the association of an Alzheimer's disease PRS with each phenotype in the three age strata using PHESANT (version 14). A description of PHESANT's automated rule-based method is published elsewhere[54]. We accounted for the multiple tests performed by generating adjusted $P$ values, controlling for a 5% false discovery rate. The threshold ($\leq 0.05$) was used as a heuristic to identify variables for follow-up in the MR analysis and not as an indicator of significance[55,56]. Categories for the ordered categorical variables are in Table 3 of the Supplementary Methods. We also estimated the effects of genetic liability to AD on previously implicated risk factors for Alzheimer's disease, selecting four factors from the Global Burden of Disease Study (high BMI, high fasting plasma glucose, smoking, and a high intake of sugar-sweetened beverages) that contributed to metrics for deaths, prevalence, years of life lost, years of life lived with disability, and disability-adjusted life-years due to AD[3]. The review identified the following as potentially modifiable risk factors for dementia; less education, midlife hypertension, obesity and hearing loss, later life smoking, depression, physical inactivity, social isolation, and diabetes. Furthermore, a meta-analysis of case-control and population-based studies showed that rheumatoid arthritis is associated with a lower incidence of Alzheimer's disease[57]. The relationship between Alzheimer's disease and rheumatoid arthritis has been studied before using genetic-based methods such as MR[57]. Hence it is not examined here. We examined the use of methotrexate (an anti-inflammatory drug for rheumatoid arthritis), due to observational studies[58,59] suggesting anti-inflammatory medicines for rheumatoid arthritis reduces the risk of Alzheimer's disease[58]. At the time of the analysis, plasma glucose was unavailable and not investigated.

**Sensitivity analysis.** We used the Nord-Trøndelag Health Study (HUNT)[60,61] to replicate the top PheWAS hits identified in the oldest age group (62–72 years) in UK Biobank, using the same ages for the stratification of the sample. The Trøndelag Health Study (HUNT) is a population-based study of ~125,000 participants, which invited the entire adult (≥20 years) population of Trøndelag[61–63]. Adults were invited for questionnaires, interviews, clinical examinations, laboratory measurements, and storage of biological samples in at least one of four study rounds so far, including HUNT1 (1984–1986, $N = 75,027$, 86.8% of invited), HUNT2 (1995–1997, $N = 65,402$, 69.7% of invited), HUNT3 (2006–2008, $N = 50,663$, 54.0% of invited), and HUNT4 (2017–2019, $N = 56,042$, 54.0% of invited)[62,63]. The current analysis includes genetic data from ~90% ($N = 71,860$) of participants from HUNT2 and HUNT3 who were genotyped by genome-wide SNP arrays in 2015[60,64]. For the replication of Alzheimer's disease PRS-outcome associations in the HUNT study, we followed up 33 outcomes (i.e., those variables available in HUNT with sufficient sample numbers for replication) that were associated with the AD PRS in UK Biobank. In the HUNT study, there is a large amount of relatedness between participants[5] therefore, to avoid the need to exclude related participants and reduce the sample size, we used a method that accounts for the genetic relatedness using a restricted maximum likelihood (REML) approach[8]. We fit a linear mixed model where a genome-wide genetic relationship matrix (GRM) was used to account for the relatedness

across the sample[60]. The models were adjusted for age, sex, and study participation round (if the outcome was measured in multiple rounds of HUNT study), batch, and ten principal components. Analyses were performed using R 4.0.3 (http://www.r-project.org) and GCTA software (version 1.93.3beta2)[8]. More details on the cohort and variable definition can be found in the Supplementary Methods. We repeated the PheWAS for the entire sample in UK Biobank without stratifying by age to maximize the power to detect associations. Furthermore, to examine if any detected associations could be attributed to the variants in or near the *APOE* gene, we repeated the PheWAS on the entire sample, excluding this region from the PRS.

## Follow-up using Mendelian randomization

We investigated whether the variables identified in our PheWAS or previously reported risk factors[3] were a cause or consequence of Alzheimer's disease using bidirectional two-sample MR (details in the Supplementary Methods). MR is a method that uses alleles randomly allocated at conception as instrumental variables to estimate the causal effect of an exposure on an outcome[5]. MR is less prone to the bias of confounding and reverse causation associated with observational studies. However, for MR to produce unbiased causal effect estimates, each genetic variant that is used as an instrumental variable must fulfill three assumptions: (1) that it is associated with the exposure (relevance assumption), (2) that it is not associated with the outcome through a confounding pathway (exchangeability assumption), and (3) is only associated with the outcome through the exposure (exclusion restriction assumption). More details on terms related to MR can be found in the MR dictionary[65].

In this MR analysis, we only considered risk factors identified by the PheWAS (in ages 62–72 years, which included the phenotypes identified in the younger age groups) and literature reviews. We identified SNPs that are strongly associated ($P \leq 5 \times 10^{-8}$) with each trait. SNPs in the *APOE* region[53] were removed from instruments proxying the exposures. For wheeze/whistling, we also examined the measured phenotype of forced vital capacity as a better measure of respiratory function. For spherical power, we derived four binary variables to indicate myopia (spherical power $< -0.5$) and hypertropia (spherical power $> 0.5$) in each eye. Exposure GWAS were based on summary statistics from UK Biobank and were performed with the BOLT-LMM software package[66] using a published pipeline[7] unless there was a larger published GWAS.

**Alzheimer's disease GWAS.** We used the same meta-analysis of the IGAP consortium[50], ADSP[51], and PGC[52] described above for the two-sample MR analyses.

**Estimating the effects of risk factors on Alzheimer's disease.** We harmonized the exposure and outcome GWAS (details in Supplementary Methods). We estimated the effect of each exposure on Alzheimer's disease using MR and the inverse-variance weighted (IVW) estimator[67]. This estimator assumes that there is no directional horizontal pleiotropy (i.e., on average, the random effects on the outcome through pathways other than the exposure are not equal to zero) and that all the genetic variants are valid instrumental variables[5]. Furthermore, IVW uses weights that treat the genetic variant-exposure associations to be known rather than estimated (i.e., the No Measurement Error assumption). When genetic variants violate the NoME assumption, causal effect estimates may exhibit weak instrument bias, estimated with the F-statistic[68,69]. The F-statistic estimates the strength of association of the genetic variant with the exposure, indicating instrument strength (the larger the F-statistic, the stronger the instrument and the larger the statistical power)[68]. We present adjusted $P$ values for inverse-variance weighted regression accounting for the number of results in the follow-up using the false discovery rate method.

**Assessing pleiotropy.** We investigated whether the SNPs had pleiotropic effects on the outcome other than through the exposure using MR Egger regression[70,71]. Egger regression allows for pleiotropic effects that are independent of the effect on the exposure of interest (InSIDE assumption)[70,72,73]. Egger regression is similar to IVW, except that it includes an intercept term representing the average pleiotropic effect. Similar to IVW, MR Egger also assumes no measurement error. To quantify the strength of the NOME violation for MR Egger, we report the $I^2G_x$ statistic[74], which indicates the expected relative bias of the MR Egger effect estimates.

**Assessing causal direction.** We used Steiger filtering to interrogate the direction of causation between Alzheimer's disease and the phenotypes identified in the PheWAS[10]. Steiger filtering assumes that a valid instrumental variable should explain more variance in the exposure (e.g., forced vital capacity) than the outcome (i.e., Alzheimer's disease), which should be true if the hypothesized direction from body mass index to Alzheimer's disease is true. We repeated MR analyses removing SNPs which explained more variance in the outcome than in the exposure.

### Reporting summary
Further information on research design is available in the Nature Research Reporting Summary linked to this article.

## Data availability
The UK Biobank Study provided the data in this study (www. ukbiobank.ac.uk), received under the data request application no. 16729. The Alzheimer's disease GWAS included IGAP, ADSP, and PGC summary statistics. Summary statistics from IGAP are publicly available at http://web.pasteur-lille.fr/en/recherche/u744/igap/igap_download.php. Summary statistics for ADSP can be obtained through a data access request https://dss.niagads.org/documentation/data-application-and-submission/application-instructions/. Summary statistics from the PGC consortium are available at https://www.med.unc.edu/pgc/download-results/. The exposure GWAS in the follow-up MR studies were performed by Ben Elsworth and are publicly available at https://gwas.mrcieu.ac.uk. The GWAS on the blood-based biomarkers was performed by Roxanna Korologou-Linden, using the UK Biobank pipeline and can be provided upon request.

## Code availability
Scripts are available on Github at https://github.com/rskl92/AD_PHEWAS_UKBIOBANK[75].

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

## Acknowledgements

This work was supported by a BRACE Alzheimer's charity (BR16/028) grant awarded to E.S., L.D.H., E.L.A., and G.D.S. It is also part of a project entitled 'social and economic consequences of health: causal inference methods and longitudinal, intergenerational data', which is part of the Health Foundation's Social and Economic Value of Health Research Programme (Award 807293). The Health Foundation is an independent charity committed to improving health and health care for people in the UK. R.K.L. was supported by a Wellcome Trust PhD studentship (Grant ref.: 215193/Z18/Z) and previously BRACE Alzheimer's charity (BR16/028). The Medical Research Council (MRC) and the University of Bristol support the MRC Integrative Epidemiology Unit [MC_UU_00011/1]. E.L.A. was funded by a UK Medical Research Council Skills Development Fellowship [MR/P014437/1]. N.M.D. is supported by an Economics and Social Research Council (ESRC) Future Research Leaders grant [ES/N000757/1] and the Norwegian Research Council Grant number 295989. L.D.H. is funded by a Career Development Award from the UK Medical Research Council (MR/M020894/1). L.A.C.M. is funded by a University of Bristol Vice-Chancellor's Fellowship. L.B. and B.M.B. receive support from the K.G. Jebsen Center for Genetic Epidemiology funded by Stiftelsen Kristian Gerhard Jebsen; Faculty of Medicine and Health Sciences, NTNU; The Liaison Committee for education, research, and innovation in Central Norway; and the Joint Research Committee between St. Olavs Hospital and the Faculty of Medicine and Health Sciences, NTNU. The PheWAS analysis was conducted using the UK Biobank Resource under Application Number 16729. The genotyping in HUNT was financed by the National Institute of Health (NIH); University of Michigan; The Research Council of Norway; The Liaison Committee for education, research and innovation in Central Norway; and the Joint Research Committee between St. Olavs Hospital and the Faculty of Medicine and Health Sciences, NTNU. The HUNT Study is a collaboration between the HUNT Research Centre (Faculty of Medicine and Health Sciences, NTNU, Norwegian University of Science and Technology), Trøndelag County Council, Central Norway Regional Health Authority, and the Norwegian Institute of Public Health. For the follow-up Mendelian randomization analyses, we used output from the MRC IEU UK Biobank GWAS pipeline. The MRC IEU UK Biobank GWAS pipeline was developed by B. Elsworth, R. Mitchell, C. Raistrick, L. Paternoster, G. Hemani, T. Gaunt (https://doi.org/10.5523/bris.pnoat8cxo0u52p6ynfaekeigi). The Alzheimer's GWAS included IGAP, ADSP, and PGC. We acknowledge the members of the Psychiatric Genomics Consortium. The Alzheimer's Disease Sequencing Project (ADSP) comprises two Alzheimer's Disease (AD) genetics consortia and three National Human Genome Research Institute (NHGRI)-funded Large Scale Sequencing and Analysis Centers (LSAC). The two AD genetics consortia are the Alzheimer's Disease Genetics Consortium (ADGC), funded by the NIA (U01 AG032984), and the Cohorts for Heart and Aging Research in Genomic Epidemiology (CHARGE), funded by the NIA (R01 AG033193), the National Heart, Lung, and Blood Institute (NHLBI), other NIH institutes, and other foreign governmental and nongovernmental organizations. The Discovery Phase analysis of sequence data is supported through UF1AG047133 (to G. Schellenberg, L.A. Farrer, M.A. Pericak-Vance, R. Mayeux, and J.L. Haines); U01AG049505 to S. Seshadri; U01AG049506 to E. Boerwinkle; U01AG049507 to E. Wijsman; and U01AG049508 to A. Goate. Data generation and harmonization in the follow-up Phases is supported by U54AG052427 (to G. Schellenberg and Wang). The ADGC cohorts include Adult Changes in Thought (ACT), the Alzheimer's Disease Centers (ADC), the Chicago Health and Aging Project (CHAP), the Memory and Aging Project (MAP), Mayo Clinic (MAYO), Mayo Parkinson's Disease controls, the University of Miami, the Multi-Institutional Research in Alzheimer's Genetic Epidemiology Study (MIRAGE), the National Cell Repository for Alzheimer's Disease (NCRAD), the National Institute on Aging Late Onset Alzheimer's Disease Family Study (NIA-LOAD), the Religious Orders Study (ROS), the Texas Alzheimer's Research and Care Consortium (TARC), Vanderbilt University/Case Western Reserve University (VAN/CWRU), the Washington Heights-Inwood Columbia Aging Project (WHICAP) and the Washington University Sequencing Project (WUSP), the Columbia University Hispanic–Estudio Familiar de Influencia Genetica de Alzheimer (EFIGUREA), the University of Toronto (UT), and Genetic Differences (GD). The CHARGE cohorts with funding provided by 5RC2HL102419 and HL105756, include the following: the Atherosclerosis Risk in Communities (ARIC) Study which is conducted as a collaborative study supported by NHLBI contracts (HHSN268201100005C, HHSN268201 100006C, HHSN268201100007C, HHSN268201100008C, HHSN268 201100009C, HHSN268201100010C, HHSN268201100011C, and HHSN268201100012C), the Austrian Stroke Prevention Study (ASPS), the Cardiovascular Health Study (CHS), the Erasmus Rucphen Family Study (ERF), the Framingham Heart Study (FHS), and the Rotterdam Study (RS). The 3 LSACs are the Human Genome Sequencing Center at the Baylor College of Medicine (U54 HG003273), the Broad Institute Genome Center (U54HG003067), and the Washington University Genome Institute (U54HG003079). Biological samples and associated phenotypic data used in primary data analyses were stored at Study Investigators institutions and at the National Cell Repository for Alzheimer's Disease (NCRAD, U24AG021886) at Indiana University, funded by the NIA. Associated Phenotypic Data used in primary and secondary data analyses were provided by Study Investigators, the NIA-funded Alzheimer's Disease Centers (ADCs), and the National Alzheimer's Coordinating Center (NACC, U01AG016976) and the National Institute on Aging Genetics of Alzheimer's Disease Data Storage Site (NIAGADS, U24AG041689) at the University of Pennsylvania, funded by the NIA and at the Database for Genotypes and Phenotypes (dbGaP) funded by the NIH. This research was partly supported by the Intramural Research Program of the NIH and the National Library of Medicine. Contributors to the Genetic Analysis Data included Study Investigators on projects that were individually funded by the NIA and other NIH institutes, and by private U.S. organizations, or foreign governmental or nongovernmental organizations. We also thank the International Genomics of Alzheimer's Project (IGAP) for providing summary results data for these analyses. The investigators within IGAP contributed to the design and implementation of IGAP and/or provided data but did not participate in the analysis or writing of this report. IGAP was made possible by the generous participation of the control subjects, the patients, and their families. The i–Select chips were funded by the French National Foundation on Alzheimer's disease and related disorders. EADI was supported by the LABEX (laboratory of excellence program investment for the future) DISTALZ grant, Inserm, Institut

Pasteur de Lille, Université de Lille 2 and the Lille University Hospital. GERAD was supported by the Medical Research Council (Grant n° 503480), Alzheimer's Research UK (Grant n° 503176), the Wellcome Trust (Grant n° 082604/2/07/Z), and German Federal Ministry of Education and Research (BMBF): Competence Network Dementia (CND) grant n° 01GI0102, 01GI0711, 01GI0420. CHARGE was partly supported by the NIH/NIA grant R01 AG033193 and the NIA AG081220 and AGES contract N01–AG–12100, the NHLBI grant R01 HL105756, the Icelandic Heart Association, and the Erasmus Medical Center and Erasmus University. ADGC was supported by the NIH/NIA grants: U01 AG032984, U24AG021886, U01AG016976, and the Alzheimer's Association grant ADGC–10–196728.

## Author contributions

R.K.L., E.L.A., N.M.D., E.S., and L.D.H. conceptualized the study; R.K.L. performed the statistical analyses in UK Biobank, and the follow-up Mendelian randomization analyses; L.M. provided the software and helped in implementing the code and troubleshooting. LB and BMB performed the statistical analyses in the HUNT study. R.K.L., L.B., B.M.B., K.K., L.D.H., D.W., Y.B.S., G.D.S., E.L.A., E.S., and N.M.D. discussed the results and provided comments on the paper.

## Competing interests

The authors declare no competing interests.
