## [Peer Review File · Nature Communications]

The causes and consequences of Alzheimer's disease:
phenome-wide evidence from Mendelian randomizationREVIEWER COMMENTS

Reviewer #1 (Remarks to the Author):

The manuscript by Korologou-Linden et al evaluates an Alzheimer's disease (AD) polygenetic risk score against approximately 15,400 traits in the UK Biobank using a PheWAS approach to understand potential factors influencing or influenced by Alzheimer's disease risk. In order to ascertain the potential causality of these traits against Alzheimer's disease risk, the authors then perform bi-directional Mendelian Randomization (MR) of these risk factor's exposures on disease outcome, and identify several traits with some evidence of causality on AD liability.

Overall, the study reads well and the approach to identifying potential risk factors that might have a causal role to play in AD pathophysiology is sound and may lead to novel interventions in the future. However, I have some comments in regards to some of the conclusions drawn with regards to the results ascertained:

The authors present association results of the PRS to all traits broken down by tertiles, and find that many of the strongest effects manifest themselves in the older group more significantly than the younger group. The authors don't delve into this observation more than attributing it to prodromal disease or ascertainment bias, but some of these traits, namely BMI, have been shown to have non-linear relationships with age at the subject level, especially in late age. These relationships should be explored more in this study, and potentially accounted for in the PheWAS, especially for the oldest tertile.

The MR results as presented reach nominal statistical significance, but after multiple-test correction, none of these results are significant. This would indicate that we cannot reject the null for any of these risk factors in regards to their causal link to AD. The authors should address this point in the discussion.

The MR was performed using GWAS results from UK Biobank. However, given that the PRS associations were observed using the UK Biobank data, the authors could additionally test these exposures from an out-of-sample study such as FinnGen, where many of the traits could potentially be mapped and for which summary statistics are available. The authors should attempt to test some of their PRS-trait association findings in this cohort, or another with similar sample size and data availability, as this might strengthen their study overall.

Overall, there appears to be significant heterogeneity in the MR estimates, potentially due to several underlying mechanisms influencing disease risk. Additionally, some of the findings in the manuscript point to specific processes, including with blood cell traits, as having strong associations with AD risk. The authors should address how one could evaluate pathway specific PRS's in the context of AD and how one would use these to inform causality of disease processes.

Minor comments and grammar:

In the Abstract background, the first sentence states "Alzheimer's disease is the leading cause of disability and healthy life years lost", but don't state in which population, or qualify this statement with more information regarding age.

Line 20: "identified to BE associated"

Line 20: sentences shouldn't start with a number; these should be spelled out

Line 31: something after PheWAS, like "can potentially"

Line 33-35: I would remove the figure links, and simply put these in the methods.

Line 46: QC usually implies that the quality of the data was not commensurate with the study, but in this case, subjects were excluded due to familial relatedness and non-Caucasian ancestry. This should be stated as such.

Supplementary Figure 4 has a green box where it looks like a forest plot should be, please check this.

Line 186: Please provide results for Steiger filtering section, not just methods link.

Reviewer #2 (Remarks to the Author):

The authors present a really comprehensive analysis of all phenotypes in the UK BioBank. Since the UKBB participants are relatively young for neurodegenerative disorders, and the age at onset of e4e4 carriers is 16 years earlier than e4 non-carriers (Liu et al 2013), it is likely that the phenotypic associations reported in the paper are attributed to the risk factors associated with APOE, but not to AD or the genetic liability to AD.

The authors report significant associations of genetic liability to AD with over 160 phenotypes. The finding looks impressive however the phenotypes are likely to be correlated. This was not clearly addressed, and the correlation between the phenotypes not shown.

They also test the AD PRS based upon 18 SNPs. Due to the strong effect of APOE and weak effects of a small number of other SNPs, it is not surprising that their results are driven by the APOE mostly. When they have run the analyses without APOE, as expected only a few phenotypes have shown association with their PRS without APOE. In addition to their sensitivity analyses, it would be interesting to see whether APOE alone can find the same associations as their PRS with APOE. If yes, then the paper should state this, and the PRS component should be downweighted, e.g. in the Abstract: "Genetic liability for Alzheimer's disease is associated with ...", should be corrected to "APOE is associated with ..."

The absence of associations in the older tertile when the APOE region was excluded from their PRS, can be potentially explained by (Fulton-Howard B. et al 2020). However, it does not imply that the PRS based on other SNPs with weaker effects is not associated (Bellou et al 2020).

In conclusion, the analyses are fantastic, the results are as expected and make sense, but the study limitations (in fact clearly stated by the authors) may lead to the results being misinterpreted by the reader, who may not pay attention to the details of the concept of "genetic liability".

REF:

1. Liu, C.-C. et al. Nat. Rev. Neurol. 9, 106–118 (2013); doi:10.1038/nrneurol.2012.263
2. Fulton-Howard, B. et al. Greater effect of polygenic risk score for Alzheimer's disease among younger cases who are apolipoprotein E-ε4 carriers. Neurobiol. Aging (2020). doi:10.1016/j.neurobiolaging.2020.09.014
3. Bellou, E. et al. Age-dependent effect of APOE and polygenic component on Alzheimer's disease. Neurobiol. Aging (2020). doi:10.1016/j.neurobiolaging.2020.04.024

Minor corrections.

Introduction: References to Figures should go to the results.

Line 94. The authors should be more specific about the "details in Supplementary material ". They can point out the exact Table/Section/page as their Supplemental material is 52 pages long.

The authors report the I2Gx and F-statistic indicating/meaning what? For the readers who are not familiar with the pleiotropy measures, it will not be clear.

"Steiger filtering examines whether the SNPs for each of the phenotypes used in the two-sample MR explain more variance in the phenotypes than in Alzheimer's disease." Not clear for non-specialists. The authors should expand this statement with an example: two-sample MR (e.g. ??) in the phenotypes (e.g. ??)

Line 123-124: Where in the supplemental material is it?

Line 148-149: "We repeated the analysis estimating the associations of the PRS and the phenotypes for the entire sample", could the authors be more specific? For all XXX phenotypes? Are the additional associated phenotypes correlated with the main ones?

Supplemental, Table 4. What is “mu”?

Reviewer #3 (Remarks to the Author):

This is a large PheWAS and Mendelian randomization (MR) study undertaken for Alzheimer’s disease (AD). The analysis incorporated commonly used approaches and many details were listed in supplementary document. One impression is that the findings are not something very new or conclusive. They may want to emphasize the novel findings from the study.

1. Page 4: Main analysis: “The full UK Biobank sample was divided into three subsamples (n=111,656 in each tertile...” They did not mention what variable was based on for this sample division. I think the division of samples was based on age as shown in the Results section. Please revise this sentence. It would be better to provide the rationale of why they wanted to divide samples in this paragraph as well.

2. The genetic variants used for computing PRS are the risk variants for late onset AD. Therefore, their PheWAS was to detect the exposures that are associated with late onset AD risk. In each age tertile, the lower age group, if there are AD subjects in this subset, it would be early onset AD. I am curious if it has any impact in PRS of AD association with each phenome in this subset if subjects with ADs in this subset are excluded.

3. In the PheWAS analysis using PRS for AD, they conducted both analyses based on PRS including and excluding APOE region. For the MR analysis (from page 5, line 80), it is not clear if they have done the MR for the scenario of excluding the APOE region. Please clarify. It would be a good secondary MR analysis for comparison if they didn’t.

4. Page 5, Follow-up using MR section: “For each risk factor identified by the PheWAS (in ages 62-72 years and literature review...” It seems that they only followed up with the risk factors identified from ages 62-72 subset. Does this mean that the rest of MR only focused on this set of risk factors? It will be better to clarify.

5. Page 8, lines 148-153, Sensitivity analysis section: this paragraph described the results of using the entire sample. However, they only mentioned those new findings. I would suggest them to describe how much overlapping findings with age-stratified analysis first for the purpose of showing sensitivity analysis results.

6. Some of their findings were in opposite directions as one would expect. For example, their MR results showed higher moderate physical activity was associated with a higher risk of AD. Before making this conclusion to public, it will be important to carefully examining the data and analysis. For example, was physical activity documented well in the dataset (UKB)? Is it possible the results due to missing adjustment of certain confounders? Can they replicate the findings using different set of GWAS summary statistics for physical activity? Since APOE has been found to associate with many other diseases or ‘phenome’, will this conclusion be biased by APOE? They cited Andrews et al. paper in their discussion (reference 47). However, this citation was from bioRxiv in 2019. The actual paper was published (PMID: 32996171). In fact, they did not find moderate-to-vigorous physical activity associated with AD in the PRS including or excluding APOE (Table 3 in their paper).

7. In the first paragraph, it would be better to summarize the 6 phenotypes found in this study first so readers can have an overall picture of the results in this study.

REVIEWER COMMENTS

Reviewer #1 (Remarks to the Author):

The manuscript by Korologou-Linden et al evaluates an Alzheimer's disease (AD) polygenetic risk score against approximately 15,400 traits in the UK Biobank using a PheWAS approach to understand potential factors influencing or influenced by Alzheimer's disease risk. In order to ascertain the potential causality of these traits against Alzheimer's disease risk, the authors then perform bi-directional Mendelian Randomization (MR) of these risk factor's exposures on disease outcome, and identify several traits with some evidence of causality on AD liability. Overall, the study reads well and the approach to identifying potential risk factors that might have a causal role to play in AD pathophysiology is sound and may lead to novel interventions in the future. However, I have some comments in regards to some of the conclusions drawn with regards to the results ascertained:

1. The authors present association results of the PRS to all traits broken down by tertiles, and find that many of the strongest effects manifest themselves in the older group more significantly than the younger group. The authors don't delve into this observation more than attributing it to prodromal disease or ascertainment bias, but some of these traits, namely BMI, have been shown to have non-linear relationships with age at the subject level, especially in late age. These relationships should be explored more in this study, and potentially accounted for in the PheWAS, especially for the oldest tertile.

In order to examine the effects of genetic liability for Alzheimer's disease on the phenome in a hypothesis-free, automated manner, we used the PHESANT package³. This tests associations using a rule-based algorithm to determine the suitable coding for each UK Biobank variable (continuous, ordered categorical, unordered categorical and binary) and the test of association to apply (linear regression, ordered logistic regression, multinomial logistic regression, and binomial logistic regression). An advantage of this approach is that we can systematically assess thousands of phenotypes very quickly and efficiently. A limitation of this approach, however, is that it is challenging to systematically use more complex modelling such as non-linear models. It is possible that BMI has a non-linear relationship with age. However, our modelling of the effects of AD liability on phenotypes was indeed non-linear, as effects were examined within tertiles of age. If there was a non-linear relationship between BMI and the AD PRS by age, our study design would, in principle, detect it. Therefore, we do not believe that further analyses of the non-linear relationship between BMI and the AD PRS are likely to change the conclusions of our paper.

2. The MR results as presented reach nominal statistical significance, but after multiple-test correction, none of these results are significant. This would indicate that we cannot reject the null for any of these risk factors in regards to their causal link to AD. The authors should address this point in the discussion. Can't reject null, but not power-related issue.

We agree with the reviewer that there is very weak evidence that any of the factors identified in the PheWAS change the risk for Alzheimer's disease, and this is indeed the very essence of the findings from our study. There is a vast evidence base of observational studies, implicating many factors in the development of Alzheimer's disease such as cholesterol, body mass index and blood-based biomarkers. Once we account for multiple testing, there is very little evidence from the MR analyses to suggest effects are likely to be causal. We added a sentence to the abstract on lines 22-28 of page 2-3:

"Genetic liability for Alzheimer's disease is associated with over 160 variables, some as early as age 39 years. These effects were largely driven by the APOE gene. Findings from MR analyses imply that most of these associations are likely to be a consequence of prodromal disease or selection, rather than the risk factor causing the disease."

We have added a paragraph on lines 321-335 of pages 15-16 in the Discussion:

"Our study shows that the most likely reason for the contrast between the primarily null MR findings in our study, and the vast evidence base for Alzheimer's disease risk factors in observational studies, is that none of the factors identified to be associated with Alzheimer's disease in such studies are likely to be causal. Rather, they are either symptoms of prodromal Alzheimer's disease (i.e. biased by reverse causation or spurious associations as a result of confounding). This conclusion is corroborated by the findings observed in the phenome-wide association study which are similar to those in observational studies (such as body mass index and sleep disturbances). Additionally, the lack of positive findings in our study was not a power-related issue, as the effects were precisely estimated. Unlike polygenic risk scores and association studies, Mendelian randomization aims to disentangle causality from correlation. Observational studies are prone to biases such as confounding and reverse causation. On the contrary, as Mendelian randomization uses genetic variants as proxies for the tested exposures, the possibility of confounding and measurement error is negligible⁴. Hence, our study highlights that most of the risk factors identified in observational literature are likely a response to Alzheimer's disease pathological processes."

3. The MR was performed using GWAS results from UK Biobank. However, given that the PRS associations were observed using the UK Biobank data, the authors could additionally test these exposures from an out-of-sample study such as FinnGen, where many of the traits could potentially be mapped and for which summary statistics are available.

We extracted SNPs for Alzheimer's disease using an out-of-sample GWAS meta-analysis (IGAP, PGC and ADSP), which does not include the UK Biobank. Hence, there is no sample overlap between the exposure (i.e. Alzheimer's disease) and outcome (i.e. UK Biobank phenome). This also applies to the reverse direction, whereby we use GWAS for the exposures from the UK Biobank, with the AD GWAS for the outcome, which does not contain UK Biobank. FinnGen has released summary statistics for smaller sample sizes (n~260,000). Hence, performing further Mendelian randomization analyses with samples that have a lower statistical power would not strengthen our study. In addition, we have now replicated the PheWAS hits in an independent cohort – the HUNT study (see detailed response to comment 4 of Reviewer 1, below).

4. The authors should attempt to test some of their PRS-trait association findings in this cohort, or another with similar sample size and data availability, as this might strengthen their study overall.

We agree that replication of our PheWAS results would indeed strengthen our study. We were unable to obtain FinnGen data (no response to emails as yet). However, we have since collaborated with researchers from the HUNT study and we were able to replicate many of our results. We have added the following paragraph on lines 91-106 of page 6 in the Methods:

“We used the Nord-Trøndelag Health Study (HUNT)¹⁹ to replicate the top PheWAS hits identified in the oldest age group (62-72 years) in UK Biobank, using the same ages for the stratification of the sample. The Trøndelag Health Study (HUNT) is a population-based study of ~125,000 participants, which invited the entire adult (≥20 years) population of Trøndelag¹⁻³. For the replication of Alzheimer disease PRS-outcome associations in the HUNT study, we followed up 32 outcomes (i.e. those variables available in HUNT with sufficient sample numbers for replication) that were found to be associated with the AD PRS in UK Biobank. In the HUNT study, there is a large amount of relatedness between participants⁵ therefore, to avoid the need to exclude related participants and reduce sample size, we used a method that accounts for the genetic relatedness using restricted maximum likelihood (REML) approach⁸. We fit a linear mixed model where a genome-wide genetic relationship matrix (GRM) was used to account for the relatedness across the sample⁸. The models were adjusted for age, sex and study participation round (if the outcome was measured in multiple rounds of HUNT study), batch, and 10 principal components. Analyses were performed using R 4.0.3 (<http://www.r-project.org>) and GCTA software⁸. More details on the cohort and variable definition can be found in the Supplement 1.”

Our results of the replication can be found on lines 205-213 of page 11 in Results:

“In participants aged 62 to 72 years, of the 165 variables identified in the UK Biobank PheWAS, we were able to replicate 32 variables with adequate precision for the age-stratified analysis, 20 of which were directionally consistent, and all were replicated at $p \leq 0.05$. The effects of genetic liability to Alzheimer's disease on blood-based biomarkers and physical measures in HUNT, closely mirror those in UK Biobank (Fig. 6). Other replicated effects included a lower odds of the participant's mother having diabetes, dietary habits such as a higher oily fish intake and fresh fruit intake, and lifestyle habits such as

frequent sleeplessness/insomnia (Supplementary Figs 4,5, 6-8). Forest plots for all measures are in in Figs 4-8, Supplement 2.”

5. Overall, there appears to be significant heterogeneity in the MR estimates, potentially due to several underlying mechanisms influencing disease risk.

We agree that there is heterogeneity in the MR estimates of the risk factors on risk of AD. One explanation for this heterogeneity is that there are multiple underlying mechanisms. However, MR and instrumental variable estimators for binary outcomes cannot assume a constant treatment effect, as such it is expected that the MR estimates, if sufficiently powered, would be heterogeneous. Using available data, it is not possible to determine whether this heterogeneity is due to pleiotropy or heterogeneous treatment effects.⁵

We have added the following to lines 341-346 of page 16 in the Discussion:

“The Alzheimer’s disease PRS may have horizontal pleiotropic effects on different traits and disorders, which can give rise to heterogeneity as observed in our study. MR and instrumental variable estimators for binary outcomes also cannot assume a constant treatment effect, as such it is expected that the MR estimates, if sufficiently powered, would be heterogeneous. With the data available, it is not possible to determine whether this heterogeneity is due to pleiotropy or heterogeneous treatment effects.”

6. Additionally, some of the findings in the manuscript point to specific processes, including with blood cell traits, as having strong associations with AD risk. The authors should address how one could evaluate pathway specific PRS’s in the context of AD and how one would use these to inform causality of disease processes.

We have included the following on lines 311-314 of page 15 in the Discussion:

“Once the function of the specific variants involved in specific biological pathways related to Alzheimer’s disease is further elucidated, pathway-specific polygenic risk scores may be used to inform causality of the biological phenotypes, identified in our PheWAS such as blood-based biomarkers.”

Minor comments and grammar:

7. In the Abstract background, the first sentence states “Alzheimer’s disease is the leading cause of disability and healthy life years lost”, but don’t state in which population, or qualify this statement with more information regarding age.

We have changed the following sentence on lines 2-3 in the Abstract on page 2:

“Alzheimer’s disease is the leading cause of disability and healthy life years lost in elderly people, worldwide.”

8. Line 20: “identified to BE associated”

. We have corrected the introduction on line 33 of page 4:

“Many risk factors and biomarkers have been identified to be associated with risk of Alzheimer’s disease”

9. Line 20: sentences shouldn’t start with a number; these should be spelled out:

On lines 33-34 of page 4 in the Introduction, we have corrected the sentence to:

“The majority of treatments (99.6%) developed to halt Alzheimer’s disease failed in phase I, II, or III trials.”

10. Line 31: something after PheWAS, like “can potentially”

We have changed the sentence on lines 44-46 on page 4 of the introduction:

“PheWAS can potentially elucidate the phenotypic consequences of Alzheimer’s disease, and critically when in the life course these effects emerge.”

11. Line 33-35: I would remove the figure links, and simply put these in the methods.

We have removed the figure links.

12. Line 46: QC usually implies that the quality of the data was not commensurate with the study, but in this case, subjects were excluded due to familial relatedness and non-Caucasian ancestry. This should be stated as such.

We have added the following on lines 58-59 of page 4 in the Methods:

“Briefly, participants were excluded due to familial relatedness and non-Caucasian ancestry.”

13. Supplementary Figure 4 has a green box where it looks like a forest plot should be, please check this.

We checked Supplementary figure 4 and we can see two age-stratified forest plots with the title “Forest plots showing effect estimates for the association between polygenic risk score including *APOE*, family history and dietary choices by age tertile.” Please let us know if this figure is not showing correctly.

14. Line 186: Please provide results for Steiger filtering section, not just methods link.

We have included a link for the Excel tables where the results from Steiger filtering can be found on lines 256-258 of page 13 in Results:

“However, the effect estimates for MR analyses retaining only the SNPs with the true hypothesized causal direction attenuated for body fat percentage, whole body fat and fat-free mass (Supplementary Tables 6, 7 and 10 in Supplement 3).”

Reviewer #2 (Remarks to the Author):

1. The authors present a really comprehensive analysis of all phenotypes in the UK BioBank. Since the UKBB participants are relatively young for neurodegenerative disorders, and the age at onset of e4e4 carriers is 16 years earlier than e4 non-carriers (Liu et al 2013), it is likely that the phenotypic associations reported in the paper are attributed to the risk factors associated with APOE, but not to AD or the genetic liability to AD.

We performed the age-stratified analysis to examine the earliest manifestations of the AD polygenic risk score, in order to understand more about aetiology of late-onset AD. Most of it was indeed driven by variants in the *APOE* gene which tag the $\epsilon 4$ allele, the single strongest predictor of AD. We do acknowledge that there may be pleiotropic effects on the outcome, which is one limitation of using polygenic risk scores. This is stated on lines 342-343 of page 16 in the Discussion:

“The Alzheimer’s disease PRS may have horizontal pleiotropic effects on different traits and disorders, which can give rise to heterogeneity as observed in our study.”

2. The authors report significant associations of genetic liability to AD with over 160 phenotypes. The finding looks impressive however the phenotypes are likely to be correlated. This was not clearly addressed, and the correlation between the phenotypes not shown.

We account for multiple testing using the false discovery rate strategy. However, this is overly conservative because the phenotypes (as the reviewer highlights) are correlated. We agree that it is important to be clear that many of the phenotypes are correlated. Thus, throughout the manuscript we have now replaced the word ‘phenotype’ with ‘variable’ to clarify that we are referring to the number of variables that we found associated with the Alzheimer’s disease polygenic risk score, rather than the number of distinct phenotypes.

On lines 6-9 of page 2 in the Abstract, we corrected the original sentence to:

“We performed a phenome-wide association study (PheWAS) of a polygenic risk score ($p \leq 5 \times 10^{-8}$) for Alzheimer’s disease with a wide range of variables in 334,968 participants of the UK Biobank, stratified by age tertiles.”

On lines 22-23 of page 2 in the Abstract, we corrected the original sentence to:

“Genetic liability for Alzheimer’s disease is associated with over 160 variables, some as early as age 39 years”

On lines 51-54 of page 4 in the Methods, we corrected the original sentence to:

“Our analysis proceeded in two steps. First, we ran a PheWAS of the Alzheimer’s disease PRS and all available variables in UK Biobank, stratifying the sample by age. Second, we followed-up all variables associated with the PRS using two-sample MR.”

On lines 214-218 of page 11 in the Results:

“We repeated the analysis estimating the associations of the PRS and the 21,849 variables in UK Biobank for the entire sample (there are additional phenotypes due to the higher occurrence of events in all participants) We identified effects of Alzheimer’s disease genetic liability on all the variables detected in the age-stratified analysis with larger precision, as well as with additional variables (Figs 9-12, Supplement 2).”

On lines 265-267 of page 13 in the Discussion:

“We found evidence that a minority of the variables identified in the PheWAS are likely to causally affect liability to Alzheimer’s disease.”

On lines 303-305 of page 14 in the Discussion:

“We found genetic risk for Alzheimer’s disease to be associated with several variables involving inflammatory pathways such as self-reported wheeze/whistling, monocyte count, and blood-based measures.”

On lines 371-372 of page 17 in the Conclusion of the manuscript:

“In this phenome-wide association study, we identified that a higher genetic liability for Alzheimer’s disease is associated with 165 variables of the 15,402 UK Biobank variables.”

3. They also test the AD PRS based upon 18 SNPs. Due to the strong effect of APOE and weak effects of a small number of other SNPs, it is not surprising that their results are driven by the APOE mostly. When they have run the analyses without APOE, as expected only a few phenotypes have shown association with their PRS without APOE. In addition to their sensitivity analyses, it would be interesting to see whether APOE alone can find the same associations as their PRS with APOE. If yes, then the paper should state this, and the PRS component should be downweighed, e.g. in the Abstract: “Genetic liability for Alzheimer’s disease is associated with ...”, should be corrected to “APOE is associated with ...”

Since we already perform and present analyses of the Alzheimer’s PRS both including and excluding APOE; additionally including results using just APOE would not add any new information to our study. Given that the effects observed in the APOE-inclusive score analyses were no longer observed in the non-APOE score is a strong indication that the APOE is the driving force of the causal effects observed on these phenotypes. We do not believe it is accurate to rephrase “genetic liability for Alzheimer’s disease” to “APOE ϵ 4 is associated with...”, as APOE ϵ 4 represents the single largest contributor to (and thus the most robust genetic instrument for) Alzheimer’s disease genetic liability.

However, it is important to acknowledge that it is the largest driver of the observed associations. We highlight that the effects observed in the PheWAS are driven by the APOE gene on lines 22-28 of pages 2-3:

“Genetic liability for Alzheimer’s disease is associated with over 160 variables, some as early as age 39 years. These effects were largely driven by the APOE gene. Findings from MR analyses imply that most of these associations are likely to be a consequence of prodromal disease or selection, rather than the risk factor causing the disease.”

We have also discussed that the effects observed were mainly driven by APOE4 on lines 272-289 of pages 13-14:

“The PheWAS using all Alzheimer’s disease SNPs (including those in the APOE gene) suggested that increased genetic liability for Alzheimer’s disease affected a diverse array of phenotypes such as medical history, brain-related phenotypes, physical measures, lifestyle, and blood-based measures. However, these effects appear to be largely driven by variation in the APOE gene, as our sensitivity analysis excluding the APOE region replicated only effects for family history of Alzheimer’s disease and some cognitive measures. These findings are in line with a recent study which showed that a higher PRS excluding APOE was particularly deleterious for age of onset of AD in APOE ϵ 4 carriers, with no evidence of such an effect in APOE ϵ 4 non-carriers⁶. Furthermore, these results are consistent with

observational studies⁷⁻¹² and studies in APOE-deficient mice¹³⁻¹⁶, which demonstrate the multifunctional role of APOE on longevity-related phenotypes such as changes in lipoprotein profiles^{13,17,18}, neurological disorders¹⁹, type II diabetes¹⁴, altered immune response¹⁵, and increased markers of oxidative stress¹⁶. Therefore, this strongly suggests that the effects of the Alzheimer's PRS on the phenome (e.g. atherosclerotic heart disease) are likely to be due to biological pathways related to APOE."

4. The absence of associations in the older tertile when the APOE region was excluded from their PRS, can be potentially explained by (Fulton-Howard B. et al 2020). However, it does not imply that the PRS based on other SNPs with weaker effects is not associated (Bellou et al 2020)

We thank the reviewer for the reference to these papers. Although these publications report interesting findings on the effects of age-stratified genetic liability to Alzheimer's disease on Alzheimer's disease risk, our sensitivity analysis excluding APOE is not age-stratified (and in addition they are asking a different research question), and as such their findings are not directly comparable to ours. We have, however, now referenced the paper by Howard et al⁶ in our discussion on lines 277-289 of page 13 in the Discussion.

"These findings are in line with a recent study which showed that a higher PRS excluding APOE was particularly deleterious for age of onset of AD in APOE ϵ 4 carriers, with no evidence of such an effect in APOE ϵ 4 non-carriers⁶."

In conclusion, the analyses are fantastic, the results are as expected and make sense, but the study limitations (in fact clearly stated by the authors) may lead to the results being misinterpreted by the reader, who may not pay attention to the details of the concept of "genetic liability".

We thank the reviewer for their positive comments and agree that it is important to convey that we are examining genetic liability to Alzheimer's disease, rather than the effects of having Alzheimer's disease. We hope that this is clear by our use of 'genetic liability to Alzheimer's disease' throughout the manuscript, but gladly accept advice on ways to make this even clearer.

Minor corrections.

Introduction: References to Figures should go to the results.

In response to reviewer 1 as well as mentioned above, we have removed the figure links.

Line 94. The authors should be more specific about the "details in Supplementary material ". They can point out the exact Table/Section/page as their Supplemental material is 52 pages long.

We acknowledge that the Supplementary material is very long and have decided to divide it into three different files; Supplements 1, 2 and 3. Supplement 1 contains the methods, Supplement 2 the results, and Supplement 3 contains the detailed results from PheWAS, as well as results from the follow-up MR for the top hits. We have made changes throughout the manuscript. For example, on lines 206-214 of page 11 in Results:

"In participants aged 62 to 72 years, of the 165 variables identified in the UK Biobank PheWAS, we were able to replicate 32 variables with adequate precision for the age-stratified analysis, 20 of which were directionally consistent and replicated at $p \leq 0.05$. The

effects of genetic liability to Alzheimer's disease on blood-based biomarkers and physical measures in HUNT, closely mirror those in UK Biobank (Fig. 6). Other replicated effects included a lower odds of the participant's mother having diabetes, dietary habits such as a higher oily fish intake and fresh fruit intake, and lifestyle habits such as frequent sleeplessness/insomnia (Supplementary Figs 4,5, 6-8). Forest plots for all measures are in in Figs 4-8, Supplement 2."

The authors report the I2Gx and F-statistic indicating/meaning what? For the readers who are not familiar with the pleiotropy measures, it will not be clear.

Due to word limit restrictions, we did not explain the basic principles of MR in detail, but we agree that it may be difficult for a non-specialist to understand this method. Thus, we have now added the following on lines 114-123 of page 7 in Methods:

"MR is a method that uses alleles which are randomly allocated at conception as instrumental variables to estimate the causal effect of an exposure on an outcome²⁰. MR is less prone to bias of confounding and reverse causation associated with observational studies. However, for MR to produce unbiased causal effect estimates, each genetic variant that is used as an instrumental variable must fulfill three assumptions: (1) that it is associated with the exposure (relevance assumption), (2) that is not associated with the outcome through a confounding pathway (exchangeability assumption), and (3) is only associated with the outcome through the exposure (exclusion restriction assumption). More details on terms related to MR can be found in the MR dictionary²¹."

We explain the F-statistic on lines 136-147 on page 8:

"We estimated the effect of each exposure on Alzheimer's disease using MR and the inverse-variance weighted (IVW) estimator²²; this estimator assumes that there is no directional horizontal pleiotropy (i.e. on average, the random effects on the outcome through pathways other than the exposure are not equal to zero) and that all the genetic variants are valid instrumental variables⁴. Furthermore, IVW uses weights that treat the genetic variant-exposure associations to be known, rather than estimated (i.e. the No Measurement Error assumption). When genetic variants violate the NoME assumption, causal effect estimates may exhibit weak instrument bias, which is estimated with the F-statistic^{23,24}. The F-statistic estimates the strength of association of the genetic variant with the exposure, indicating instrument strength (the larger the F-statistic, the stronger the instrument and the larger the statistical power)²³."

Additionally, we explain briefly the pleiotropy tests on lines 150-157 of page 8 in Methods: *"We investigated whether the SNPs had pleiotropic effects on the outcome other than through the exposure using MR Egger regression^{25,26}. Egger regression allows for pleiotropic effects that are independent of the effect on the exposure of interest (InSIDE assumption)^{25,27,28}. Egger regression is similar to IVW, except that it includes an intercept term which represents the average pleiotropic effect. Similarly to IVW, MR Egger also assumes no measurement error. To quantify the strength of the NOME violation for MR Egger, we report the $I^2 G_x$ statistic²⁹, which indicates the expected relative bias of the MR Egger effect estimates."*

“Steiger filtering examines whether the SNPs for each of the phenotypes used in the two-sample MR explain more variance in the phenotypes than in Alzheimer’s disease.” Not clear for non-specialists.

The authors should expand this statement with an example: two-sample MR (e.g. ??) in the phenotypes (e.g. ??)

We have now clarified the sentence on lines 159-165 of pages 8-9 in Methods:

“We used Steiger filtering to interrogate the direction of causation between Alzheimer’s disease and the phenotypes identified in the genome-wide association study³⁰. Steiger filtering assumes that a valid instrumental variable should explain more variance in the exposure (e.g. forced vital capacity) than the outcome (i.e. Alzheimer’s disease), which should be true if the hypothesized direction from body mass index to Alzheimer’s disease is true.”

Line 123-124: Where in the supplemental material is it?

We have clarified the location of these results in the sentence on lines 181 of Results on page 10:

“Selected PheWAS hits are presented in Figs 3-5 and Figs 1-3 in the Supplement 2.”

Line 148-148: “We repeated the analysis estimating the associations of the PRS and the phenotypes for the entire sample”, could the authors be more specific? For all XXX phenotypes? Are the additional associated phenotypes correlated with the main ones?

We have now added the following on lines 214-218 of page 11 in the Methods:

“We repeated the analysis estimating the associations of the PRS and the 21,849 variables in UK Biobank for the entire sample (there are additional phenotypes due to the higher occurrence of events in all participants). We identified effects of Alzheimer’s disease genetic liability on the variables detected in the age-stratified analysis with larger precision, as well as additional variables (Figs 9-12, Supplement 2).”

Supplemental, Table 4. What is “mu”?

We thank the reviewer for the comment. Mu represents the case fraction. To clarify this, we have replaced mu with the header “Case fraction”.

Reviewer #3 (Remarks to the Author):

1. This is a large PheWAS and Mendelian randomization (MR) study undertaken for Alzheimer's disease (AD). The analysis incorporated commonly used approaches and many details were listed in supplementary document. One impression is that the findings are not something very new or conclusive. They may want to emphasize the novel findings from the study.

To our knowledge, this is the first phenome-wide association study of Alzheimer's disease and the age-stratified nature of the study allowed to examine important age-varying effects. While we observed the genetic liability for Alzheimer's disease to be associated with many variables in old age, very few were replicated in the follow-up MR study. This provides evidence that most risk factors reported to be associated with Alzheimer's disease risk in observational studies, are unlikely to be causal, and more likely to be symptoms of prodromal Alzheimer's disease or spurious associations due to confounding.

2. Page 4: Main analysis: "The full UK Biobank sample was divided into three subsamples (n=111,656 in each tertile..." They did not mention what variable was based on for this sample division. I think the division of samples was based on age as shown in the Results section. Please revise this sentence. It would be better to provide the rationale of why they wanted to divide samples in this paragraph as well.

We agree that this needed clarification and have amended the sentence accordingly on lines 71-74 of page 5 in the Methods:

"The full UK Biobank sample was divided into three age-stratified subsamples (n=111,656 in each tertile), with the aim of examining the age-varying effects of the PRS for Alzheimer's disease."

3. The genetic variants used for computing PRS are the risk variants for late onset AD. Therefore, their PheWAS was to detect the exposures that are associated with late onset AD risk. In each age tertile, the lower age group, if there are AD subjects in this subset, it would be early onset AD. I am curious if it has any impact in PRS of AD association with each phenome in this subset if subjects with ADs in this subset are excluded.

To our knowledge, less than 1% of the UK Biobank sample have any form of dementia, which is a very small proportion of the total sample (N=500,000+). Furthermore, polygenic risk scores are not deterministic as they represent underlying liability. The exclusion of Alzheimer's disease cases would be removing the participants that we would be most interested in capturing.

4. In the PheWAS analysis using PRS for AD, they conducted both analyses based on PRS including and excluding APOE region. For the MR analysis (from page 5, line 80), it is not clear if they have done the MR for the scenario of excluding the APOE region. Please clarify. It would be a good secondary MR analysis for comparison if they didn't.

In the manuscript on lines 124-127 of page 7, we have written:

"In this MR analysis, we only considered risk factors identified by the PheWAS (in ages 62-72 years, which included the phenotypes identified in the younger age groups) and literature reviews and identified SNPs that are strongly associated ($p \leq 5 \times 10^{-8}$) with each trait. SNPs in the APOE region³¹ were removed from instruments proxying the exposures."

One of the assumptions of Mendelian randomization is exclusion restriction: that the genetic variant affects the outcome, conditional on the exposure. *APOE* is the greatest genetic risk factor for Alzheimer's disease. Hence, if *APOE* was included as a genetic instrument for the exposure, it is likely we would introduce bias due to horizontal pleiotropy.

5. Page 5, Follow-up using MR section: "For each risk factor identified by the PheWAS (in ages 62-72 years and literature review....". It seems that they only followed up with the risk factors identified from ages 62-72 subset. Does this mean that the rest of MR only focused on this set of risk factors? It will be better to clarify.

Most of the effects of AD genetic liability were observed in the oldest age group and the variables that were detected in the earlier age groups, remained in the oldest age group (age 62-72 years). On lines 124-126 of page 7, we changed the sentence to:

"In this MR analysis, we only considered risk factors identified by the PheWAS (in ages 62-72 years, which included the phenotypes identified in the younger age groups) and literature reviews and identified SNPs that are strongly associated ($p \leq 5 \times 10^{-8}$) with each trait."

6. Page 8, lines 148-153, Sensitivity analysis section: this paragraph described the results of using the entire sample. However, they only mentioned those new findings. I would suggest them to describe how much overlapping findings with age-stratified analysis first for the purpose of showing sensitivity analysis results.

We agree and have added the following sentences on lines 214-218 of page 11 in the Results:

"We repeated the analysis estimating the associations of the PRS and the 21,849 variables in UK Biobank for the entire sample (there are additional phenotypes due to the higher occurrence of events in all participants). We identified effects of Alzheimer's disease genetic liability on all the variables detected in the age-stratified analysis with larger precision, as well as additional variables (Figs 9-12, Supplement 2)."

7. Some of their findings were in opposite directions as one would expect. For example, their MR results showed higher moderate physical activity was associated with a higher risk of AD. Before making this conclusion to public, it will be important to carefully examining the data and analysis.

We agree that it is important to ensure that epidemiological results are interpreted correctly, so that no misleading conclusions are conveyed to the public. We have double checked the data and we are confident that the result is correct. The data is publicly available, and our code is available on github (link to github page provided in methods), so that findings can be replicated. It is also directionally consistent with previously published data². We have discussed associations which are contrary to what may have expected *a priori*, and we have given several potential explanations for these results. We have added an additional line stating clearly that this result should be interpreted with caution. This paragraph can be found on lines 348-368 on page 16:

“The results from both the PheWAS and the MR follow-up could be explained by collider bias, which may have been introduced into our study through selection of the study sample. The UK Biobank includes a highly selected, healthier sample of the UK population³². Compared to the general population, participants are less likely to be obese, to smoke, to drink alcohol daily, and had fewer self-reported medical conditions³³. Selection bias may occur if those with a lower genetic liability to Alzheimer’s disease and a specific trait (e.g. higher education or higher levels of physical activity) are more likely to participate in the study. This could induce an association between genetic risk for Alzheimer’s disease and the traits in our study³⁴. Furthermore, if both the PRS for Alzheimer’s disease and the examined traits associate with survival, sampling only living people can induce spurious associations that do not exist in the general population^{35,36}. Such bias may have affected our findings for body mass and physical activity, as individuals with a higher body mass index or infrequent physical activity and those with higher values of the Alzheimer’s PRS are less likely to survive and participate in UK Biobank. The PRS for Alzheimer’s disease in our analysis was associated with lower age at recruitment, suggesting that older people with higher values of the score are less likely to participate. Hence, the variables that may be associated with selection or survival⁷⁰ should be interpreted with caution, considering these limitations.”

8. For example, was physical activity documented well in the dataset (UKB)? Is it possible the results due to missing adjustment of certain confounders?

Our results are extremely unlikely to be due to missing adjustment of certain confounders, as confounding is not an issue in this study, due to the use of genetic variants as the exposure²⁰. Additionally, we performed follow-up two sample MR on the phenotypes identified in the phenome-wide association study. The detected variable was self-reported moderate physical activity. There is likely to be measurement error for the phenotypic variable, but MR studies are not prone to measurement error of the exposure, in the same manner that observational studies are (because we are using genetic variants, which are measured with very little error, as the exposure). Selection bias or survival bias are, however, possible explanations for our results (as stated in the discussion paragraph above).

9. Can they replicate the findings using different set of GWAS summary statistics for physical activity?

The aim of our paper was to identify age-dependent effects of Alzheimer’s disease genetic liability and to follow-up the phenotypes using MR, exactly as they were defined in the phenome-wide association study. Objective measures of physical activity (i.e. measured through an accelerometer) are not strongly correlated with subjective self-reported measures of physical activity³⁷. However, we have since examined the effect of self-reported moderate-to-vigorous physical activity using the GWAS by Klimentidis et al, which used data from questionnaires asking participants the question “In a typical week, on how many days did you do 10 minutes or more of moderate physical activities like carrying light loads, cycling at normal pace? (Do not include walking)”. For each of these questions, those who answered 1 or more such days were then asked “How many minutes did you usually spend doing moderate/vigorous activities on a typical day”. Moderate-to-vigorous physical activity was estimated by taking the sum of total minutes/week of moderate physical activity multiplied by four and the total number of vigorous activity minutes/week multiplied by eight, reflecting their metabolic equivalents¹. Using that GWAS, we identified an odds ratio of 1.22 (95% CI: 0.59, 2.50), which is consistent with the effect estimates in our study (OR: 2.69; 95% CI: 1.39, 5.17). Please note that the exposure GWAS of moderate-to-vigorous physical

activity by Klimentidis et al is on the per category increase scale, whereas the GWAS used in our study was on the standard deviation scale.

10. Since *APOE* has been found to associate with many other diseases or ‘phenome’, will this conclusion be biased by *APOE*?

As *APOE* is the largest genetic risk factor for Alzheimer’s disease and is indeed pleiotropic, we performed all MR analyses (i.e. in the direction from risk factor TO Alzheimer’s disease risk) excluding variants in the *APOE* region, in order to avoid violating a key MR assumption (i.e. *APOE* has a direct effect on Alzheimer’s disease independent of the exposure).

11. They cited Andrews et al. paper in their discussion (reference 47). However, this citation was from bioRxiv in 2019. The actual paper was published (PMID: 32996171). In fact, they did not find moderate-to-vigorous physical activity associated with AD in the PRS including or excluding *APOE* (Table 3 in their paper).

We thank the reviewer for this comment. In our manuscript, we referred to the first version of the manuscript on bioRxiv by Andrews et al³⁸, which found an association between self-reported moderate physical activity and Alzheimer’s disease (OR: 2.5; 95% CI: 1.47, 4.23)³⁸. We have updated the reference with the published version, which reports directionally consistent findings to ours². They identified that a higher polygenic risk score for moderate to vigorous physical activity (at the 5×10^{-6}) was associated with Alzheimer’s disease (OR: 1.19; 95% CI: 1.03, 1.71), but not at the genome-wide significant threshold ($p \leq 5 \times 10^{-8}$) (OR: 1.19, 95% CIs: 0.72, 1.93). Contrary to Andrews et al², who use the GWAS of moderate-to-physical activity by Klimentidis et al, we used the GWAS run internally on the Integrative Epidemiology Unit (IEU) pipeline for the responses to the question “In a typical WEEK, on how many days did you do 10 minutes or more of moderate physical activities like carrying light loads, cycling at normal pace? (Do not include walking)”. The physical activity measure in the GWAS by Klimentidis et al¹ was weighted by the metabolic equivalent. However, the scope of our paper was to test the phenotypes using MR, exactly as they were defined in the phenome-wide association study and hence, we have not changed the exposure GWAS for the MR.

12. In the first paragraph, it would be better to summarize the 6 phenotypes found in this study first so readers can have an overall picture of the results in this study.

As per the recommendation, we have added the following sentences on lines 265-271 of page 13 in the Discussion:

“We found evidence that a minority of the variables identified in the PheWAS are likely to causally affect liability to Alzheimer’s disease. Of the variables associated with Alzheimer’s disease genetic liability in the PheWAS, these included basal metabolic rate, forced vital capacity, whole body fat-free and whole body water mass, as well as self-reported moderate physical activity. Of the factors to have previously been implicated in Alzheimer’s disease risk, these included A level/AS qualifications and college degree qualifications (Fig 6).”

References

1. Klimentidis YC, Raichlen DA, Bea J, et al. Genome-wide association study of habitual physical activity in over 377,000 UK Biobank participants identifies multiple variants including CADM2 and APOE. *International Journal of Obesity*. 2018;42(6):1161-1176.
2. Andrews SJ, Fulton-Howard B, O'Reilly P, Marcora E, Goate AM, collaborators of the Alzheimer's Disease Genetics Consortium. Causal Associations Between Modifiable Risk Factors and the Alzheimer's Phenome. *Annals of Neurology*. 2021;89(1):54-65.
3. Millard LAC, Davies NM, Gaunt TR, Davey Smith G, Tilling K. Software Application Profile: PHESANT: a tool for performing automated phenome scans in UK Biobank. *International Journal of Epidemiology*. 2018;47(1):29-35.
4. Davey Smith G, Hemani G. Mendelian randomization: genetic anchors for causal inference in epidemiological studies. *Human Molecular Genetics*. 2014;23(R1):R89-98.
5. Hartwig FP, Bowden J, Wang L, Smith GD, Davies NM. Average causal effect estimation via instrumental variables: the no simultaneous heterogeneity assumption. Published online October 20, 2020.
6. Fulton-Howard B, Goate AM, Adelson RP, et al. Greater effect of polygenic risk score for Alzheimer's disease among younger cases who are apolipoprotein E- ϵ 4 carriers. *Neurobiol Aging*. 2021;99:101.e1-101.e9.
7. Lehtinen S, Lehtimäki T, Sisto T, et al. Apolipoprotein E polymorphism, serum lipids, myocardial infarction and severity of angiographically verified coronary artery disease in men and women. *Atherosclerosis*. 1995;114(1):83-91.
8. Muros M, Rodríguez-Ferrer C. Apolipoprotein E polymorphism influence on lipids, apolipoproteins and Lp(a) in a Spanish population underexpressing apo E4. *Atherosclerosis*. 1996;121(1):13-21.
9. Khan TA, Shah T, Prieto D, et al. Apolipoprotein E genotype, cardiovascular biomarkers and risk of stroke: Systematic review and meta-analysis of 14 015 stroke cases and pooled analysis of primary biomarker data from up to 60 883 individuals. *International Journal of Epidemiology*. 2013;42(2):475-492.
10. Kulminski AM, Loika Y, Culminkaya I, et al. Independent associations of TOMM40 and APOE variants with body mass index. *Aging Cell*. 2019;18(1).
11. Eichner JE, Dunn ST, Perveen G, Thompson DM, Stewart KE, Stroehla BC. Apolipoprotein E polymorphism and cardiovascular disease: A HuGE review. *American Journal of Epidemiology*. 2002;155(6):487-495.
12. Rasmussen KL. Plasma levels of apolipoprotein E, APOE genotype and risk of dementia and ischemic heart disease: A review. *Atherosclerosis*. 2016;255:145-155.
13. Zhang SH, Reddick RL, Piedrahita JA, Maeda N. Spontaneous hypercholesterolemia and arterial lesions in mice lacking apolipoprotein E. *Science (1979)*. 1992;258(5081):468-471.
14. Li J, Wang Q, Chai W, Chen MH, Liu Z, Shi W. Hyperglycemia in apolipoprotein E-deficient mouse strains with different atherosclerosis susceptibility. *Cardiovascular Diabetology*. 2011;10(117).
15. Roselaar SE, Daugherty A. Apolipoprotein E-deficient mice have impaired innate immune responses to *Listeria monocytogenes* in vivo. *J Lipid Res*. 1998;39(9):1740-1743. <http://www.ncbi.nlm.nih.gov/pubmed/9741685>

16. Hayek T, Oiknine J, Brook JG, Aviram M. Increased plasma and lipoprotein lipid peroxidation in apo E-deficient mice. *Biochemical and Biophysical Research Communications*. 1994;201(3):1567-1574.
17. Moghadasian MH, McManus BM, Nguyen LB, et al. Pathophysiology of apolipoprotein E deficiency in mice: relevance to apo E-related disorders in humans. *The FASEB Journal*. 2001;15(14):2623-2630.
18. Plump AS, Smith JD, Hayek T, et al. Severe hypercholesterolemia and atherosclerosis in apolipoprotein E-deficient mice created by homologous recombination in ES cells. *Cell*. 1992;71(2):343-353.
19. Roses AD. Apolipoprotein E alleles as risk factors in Alzheimer's disease. *Annual Review of Medicine*. 1996;47(1):387-400.
20. Davey Smith G, Hemani G. Mendelian randomization: genetic anchors for causal inference in epidemiological studies. *Hum Mol Genet*. 2014;23(R1):R89-98.
21. Lawlor D, Wade K, Borges M, et al. A Mendelian Randomization dictionary: Useful definitions and descriptions for undertaking, understanding and interpreting Mendelian Randomization studies. Published online March 2019.
22. Burgess S, Butterworth A, Thompson SG. Mendelian randomization analysis with multiple genetic variants using summarized data. *Genet Epidemiol*. 2013;37(7):658-665.
23. Burgess S, Thompson SG. Avoiding bias from weak instruments in mendelian randomization studies. *International Journal of Epidemiology*. 2011;40(3):755-764.
24. Bowden J, Davey Smith G, Haycock PC, Burgess S. Consistent Estimation in Mendelian Randomization with Some Invalid Instruments Using a Weighted Median Estimator. *Genetic Epidemiology*. 2016;40(4):304-314.
25. Bowden J, Davey Smith G, Burgess S. Mendelian randomization with invalid instruments: Effect estimation and bias detection through Egger regression. *International Journal of Epidemiology*. 2015;44(2):512-525.
26. Egger M, Davey Smith G, Schneider M, Minder C. Bias in meta-analysis detected by a simple, graphical test. *BMJ*. 1997;315(7109):629-634.
27. Rees JMB, Wood AM, Burgess S. Extending the MR-Egger method for multivariable Mendelian randomization to correct for both measured and unmeasured pleiotropy. *Statistics in Medicine*. 2017;36(29):4705-4718.
28. Burgess S, Thompson SG. Interpreting findings from Mendelian randomization using the MR-Egger method. *European Journal of Epidemiology*. 2017;32(5):377-389.
29. Bowden J, Fabiola Del Greco M, Minelli C, Davey Smith G, Sheehan NA, Thompson JR. Assessing the suitability of summary data for two-sample mendelian randomization analyses using MR-Egger regression: The role of the I² statistic. *International Journal of Epidemiology*. 2016;45(6):1961-1974.
30. Hemani G, Tilling K, Davey Smith G. Orienting the causal relationship between imprecisely measured traits using GWAS summary data. *PLoS Genetics*. 2017;13(11).
31. Escott-Price V, Shoai M, Pither R, Williams J, Hardy J. Polygenic score prediction captures nearly all common genetic risk for Alzheimer's disease. *Neurobiology of Aging*. 2017;49:214.e7-214.e11.
32. Hughes RA, Davies NM, Davey Smith G, Tilling K. Selection Bias When Estimating Average Treatment Effects Using One-sample Instrumental Variable Analysis. *Epidemiology*. 2019;30:350-357.

33. Fry A, Littlejohns TJ, Sudlow C, et al. Comparison of Sociodemographic and Health-Related Characteristics of UK Biobank Participants with Those of the General Population. *American Journal of Epidemiology*. 2017;186(9):1026-1034.
34. Munafò MR, Tilling K, Taylor AE, Evans DM, Davey Smith G. Collider scope: When selection bias can substantially influence observed associations. *International Journal of Epidemiology*. 2018;47(1):226-235.
35. Hernán M a, Alonso A, Logroscino G. Cigarette smoking and dementia: potential selection bias in the elderly. *Epidemiology*. 2008;19(3):448-450.
36. Smit RAJ, Trompet S, Dekkers OM, Jukema JW, Cessie S. Survival Bias in Mendelian Randomization Studies. 2019;30(6):813-816.
37. Luo J, Lee RYW. Opposing patterns in self-reported and measured physical activity levels in middle-aged adults. *European Journal of Ageing*. Published online 2021.
38. Andrews SJ, Marcora E, Goate A. Causal associations between potentially modifiable risk factors and the Alzheimer's phenome: A Mendelian randomization study. *bioRxiv*. Published online January 1, 2019.

REVIEWERS' COMMENTS

Reviewer #1 (Remarks to the Author):

Thank you to the authors for addressing my comments and concerns. Based on their additional explanations/edits, and additional work to replicate their findings in another well powered cohort (HUNT), I believe this manuscript is now ready for publication, and I have no additional concerns or comments.

Reviewer #2 (Remarks to the Author):

The authors have addressed all my comments.

Reviewer #3 (Remarks to the Author):

The authors have addressed my previous questions. After reading the paper again, it will be great if authors can add some explanation for the study design.

The study design of this paper was first to use PheWAS approach to test if AD PRS is associated with any phenome. The authors then used the 'significant' phenome to investigate if they have causal effect on AD through MR approach. Based on the statistical model in PheWAS analysis, they tested the AD PRS effect on each phenome. That is AD -> phenome relationship. However, MR analysis hypothesized genetic instrument -> phenome -> AD, a different direction between phenome and AD. The author may want to explain more why it is reasonable to flip the direction between AD and phenome in these two approaches in Introduction section. Perhaps, adding some explanation that the PheWAS approach is a reverse regression model to test the phenome association with AD (as a AD risk factor).

Second, the number of variants used in AD PRS was not mentioned except in the Supplementary materials. They may want to consider to add it to Page 5, 'Polygenic Risk Score' section, by mentioning that 18 variants for AD were used to compute PRS.

Reviewer #1 (Remarks to the Author):

Thank you to the authors for addressing my comments and concerns. Based on their additional explanations/edits, and additional work to replicate their findings in another well powered cohort (HUNT), I believe this manuscript is now ready for publication, and I have no additional concerns or comments.

We thank you for your time to reviewing the manuscript and providing feedback that have considerably improved its quality and readiness for publication.

Reviewer #2 (Remarks to the Author):

The authors have addressed all my comments.

We thank you for your time in reviewing the manuscript and providing feedback that have considerably improved its quality and readiness for publication.

Reviewer #3 (Remarks to the Author):

The authors have addressed my previous questions. After reading the paper again, it will be great if authors can add some explanation for the study design.

The study design of this paper was first to use PheWAS approach to test if AD PRS is associated with any phenome. The authors then used the 'significant' phenome to investigate if they have causal effect on AD through MR approach. Based on the statistical model in PheWAS analysis, they tested the AD PRS effect on each phenome. That is AD -> phenome relationship. However, MR analysis hypothesized genetic instrument -> phenome -> AD, a different direction between phenome and AD. The author may want to explain more why it is reasonable to flip the direction between AD and phenome in these two approaches in Introduction section. Perhaps, adding some explanation that the PheWAS approach is a reverse regression model to test the phenome association with AD (as a AD risk factor).

We have added a few sentences on lines 31-35 of the Introduction to clarify the aims:

"Here, we estimated the associations of genetic liability for Alzheimer's disease and the phenome by age to identify the earliest effects of the disease process (i.e. genetic liability to Alzheimer's disease as the exposure). We then tested whether the identified variables were a cause or a consequence of Alzheimer's disease using two sample MR (i.e. phenome associated with Alzheimer's disease genetic liability as the exposure.)"

Second, the number of variants used in AD PRS was not mentioned except in the Supplementary materials. They may want to consider to add it to Page 5, 'Polygenic Risk Score' section, by mentioning that 18 variants for AD were used to compute PRS.

We have added the following on lines 46-48 of the Results:

"We examined the effects of Alzheimer's disease on the UK Biobank phenome, using 18 single nucleotide polymorphisms (SNPs) robustly and independently associated with Alzheimer's disease ($p \leq 5 \times 10^{-8}$)."

We have also added on lines 271-273 of the Methods:

“Each score was calculated from the effect size (logarithm (log) odds)-weighted sum of 18 alleles associated with Alzheimer’s disease within each participant (Supplementary Information Table 1)”